# Calibrate and Debias Layer-wise Sampling for Graph Convolutional Networks

**Yifan Chen**[1]* **Tianning Xu**[1]* **Dilek Hakkani-Tur**[2] **Di Jin**[2] **Yun Yang**[1] **Ruoqing Zhu**[1]

[1] *University of Illinois Urbana-Champaign*    [2] *Amazon Alexa AI*

{yifanc10, tx8, yy84, rqzhu}@illinois.edu    {hakkanit, djinamzn}@amazon.com

**Reviewed on OpenReview:** *https://openreview.net/forum?id=JyKNuoZGux*

## Abstract

Multiple sampling-based methods have been developed for approximating and accelerating node embedding aggregation in graph convolutional networks (GCNs) training. Among them, a layer-wise approach recursively performs importance sampling to select neighbors jointly for existing nodes in each layer. This paper revisits the approach from a matrix approximation perspective, and identifies two issues in the existing layer-wise sampling methods: suboptimal sampling probabilities and estimation biases induced by sampling without replacement. To address these issues, we accordingly propose two remedies: a new principle for constructing sampling probabilities and an efficient debiasing algorithm. The improvements are demonstrated by extensive analyses of estimation variance and experiments on common benchmarks. Code and algorithm implementations are publicly available at *https://github.com/ychen-stat-ml/GCN-layer-wise-sampling*.

## 1 Introduction

Graph Convolutional Networks (Kipf & Welling, 2017) are popular methods for learning node representations. However, it is computationally challenging to train a GCN over large-scale graphs due to the inter-dependence of nodes in a graph. In mini-batch gradient descent training for an $L$-layer GCN, the computation of embeddings involves not only the nodes in the batch but also their $L$-hop neighbors, which is known as the phenomenon of "neighbor explosion" (Zeng et al., 2020) or "neighbor expansion" (Chen et al., 2018a; Huang et al., 2018). To alleviate such a computation issue for large-scale graphs, sampling-based methods are proposed to accelerate the training and reduce the memory cost. These approaches can be categorized as node-wise sampling approaches (Hamilton et al., 2017; Chen et al., 2018a), subgraph sampling approaches (Zeng et al., 2020; Chiang et al., 2019; Cong et al., 2020), and layer-wise sampling approaches (Chen et al., 2018b; Huang et al., 2018; Zou et al., 2019). We focus on layer-wise sampling in this work, which enjoys the efficiency and variance reduction by sampling columns of renormalized Laplacian matrix in each layer.

This paper revisits the existing sampling schemes in layer-wise sampling methods. We identify two potential drawbacks in the common practice of layer-wise sampling, especially on FastGCN (Chen et al., 2018b) and LADIES (Zou et al., 2019). First, the sampling probabilities are suboptimal since a convenient while unguaranteed assumption fails to hold on many common graph benchmarks, such as Reddit (Hamilton et al., 2017) and OGB (Hu et al., 2020). Secondly, the previous implementations of layer-wise sampling methods perform sampling *without* replacement, which deviates from their theoretical results, and introduce biases in the estimation. Realizing the two issues, we accordingly propose the remedies with a new principle to construct sampling probabilities and a debiasing algorithm, as well as the variance analyses for the two propositions.

---

\* Equal contribution. The majority of this work was done prior to the first author's internship at Amazon Alexa AI.

To the best of our knowledge, our paper is the first to recognize and resolve these two issues, importance sampling assumption and the practical sampling implementation, for layer-wise sampling on GCNs. Specifically, we first investigate the distributions of embedding and weight matrices in GCNs and propose a more conservative principle to construct importance sampling probabilities, which leverages the Principle of Maximum Entropy. Secondly, we recognize the bias induced by sampling without replacement and suggest a debiasing algorithm supported by theoretical analysis, which closes the gap between theory and practice. We demonstrate the improvement of our sampling method by evaluating both matrix approximation error and the model prediction accuracy on common benchmarks. With our proposed remedies, GCNs consistently converge faster in training. We believe our proposed debiasing method can be further adapted to more general machine learning tasks involving importance sampling without replacement, and we discuss the prospective applications in Section 7.

### 1.1 Background and Related Work

**GCN.** Graph Convolutional Networks (Kipf & Welling, 2017), as the name suggests, effectively incorporate the technique of convolution filter into the graph domain (Wu et al., 2020; Bronstein et al., 2017). GCN has achieved great success in learning tasks such as node classification and link prediction, with applications ranging from recommender systems (Ying et al., 2018), traffic prediction (Cui et al., 2019; Rahimi et al., 2018), and knowledge graphs (Schlichtkrull et al., 2018). Its mechanism will be detailed shortly in Section 2.1.

**Sampling-based GCN Training.** To name a few of sampling schemes, GraphSAGE (Hamilton et al., 2017) first introduces the "node-wise" neighbor sampling scheme, where a fixed number of neighbors are uniformly and independently sampled for each node involved, across every layer. To reduce variance in node-wise sampling, VR-GCN (Chen et al., 2018a) applies a control variate based algorithm using historical activation. Instead of sampling for each node separately, "layer-wise" sampling is a more collective approach: the neighbors are jointly sampled for all the existing nodes in each layer. FastGCN (Chen et al., 2018b) first introduces this scheme with importance sampling. AS-GCN (Huang et al., 2018) proposes an alternative sampling method which approximates the hidden layer to help estimate the probabilities in sampling procedures. Then, Zou et al. (2019) propose a layer-dependent importance sampling scheme (LADIES) to further reduce the variance in training, and aim to alleviate the issue of sparse connection (empty rows in the sampled adjacency matrix) in FastGCN. In addition, for "subgraph" approaches, ClusterGCN (Chiang et al., 2019) samples a dense subgraph associated with the nodes in a mini batch by graph clustering algorithm; GraphSAINT (Zeng et al., 2020) introduces normalization and variance reduction in subgraph sampling.

To provide a more scalable improvement on sampling-based GCN training, we focus on *history-oblivious* layer-wise sampling methods (e.g., FastGCN and LADIES), which do not rely on history information to construct the sampling probabilities. Note that though some other sampling based methods, such as VR-GCN and AS-GCN, enjoy attractive approximation accuracy by storing and leveraging historical information of model hidden states, they introduce large time and space cost. For example, the training time of AS-GCN can be "even longer than vanilla GCN" (Zeng et al., 2020). Moreover, they cannot perform sampling and training separately due to the dependence of sampling probabilities on the training information.

**Debiasing algorithms for weighted random sampling.** In practice, layer-wise sampling is performed by a sequential procedure named as weighted random sampling (WRS) (Efraimidis & Spirakis, 2006), which realizes "sampling without replacement" while induces bias (analyzed in Section 5). Similar phenomena have been noticed by some studies on stochastic gradient estimators (Liang et al., 2018; Liu et al., 2019; Kool et al., 2020), which involve WRS as well; some debiasing algorithms are accordingly developed in those works. In Section 5, we discuss the issues of directly applying existing algorithms to layer-wise sampling and propose a more time-efficient debiasing method.

## 2 Notations and Preliminaries

We introduce necessary notations and backgrounds of GCNs and layer-wise sampling in this section. Debiasing-related preliminaries are deferred to Section 5.

## 2.1 Graph Convolutional Networks

The GCN architecture for semi-supervised node classification is introduced by Kipf & Welling (2017). Suppose we have an undirected graph $\mathcal{G} = (\mathcal{V}, \mathcal{E})$, where $\mathcal{V}$ is the set of $n$ nodes and $\mathcal{E}$ is the set of $E$ edges. Denote node $i$ in $\mathcal{V}$ as $v_i$, where $i \in [n]$ is the index of nodes in the graph and $[n]$ denotes the set $\{1, 2, \ldots, n\}$. Each node $v_i \in \mathcal{V}$ is associated with a feature vector $x_i \in \mathbb{R}^p$ and a label vector $y_i \in \mathbb{R}^q$. In a transductive setting, although we have access to the feature of every node in $\mathcal{V}$ and every edge in $\mathcal{E}$, i.e. the $n \times n$ adjacency matrix $A$, we can only observe the label of partial nodes $\mathcal{V}_{train} \subset \mathcal{V}$; predicting the labels of the rest nodes in $\mathcal{V} - \mathcal{V}_{train}$ therefore becomes a semi-supervised learning task.

A graph convolution layer is defined as:

$$\boldsymbol{Z}^{(l+1)} = \boldsymbol{P}\boldsymbol{H}^{(l)}\boldsymbol{W}^{(l)}, \quad \boldsymbol{H}^{(l)} = \sigma(\boldsymbol{Z}^{(l)}), \tag{1}$$

where $\sigma$ is an activation function and $\boldsymbol{P}$ is obtained from normalizing the graph adjacency matrix $\boldsymbol{A}$; $\boldsymbol{H}^{(l)}$ is the embedding matrix of the graph nodes in the $l$-th layer, and $\boldsymbol{W}^{(l)}$ is the weight matrix of the same layer. In particular, $\boldsymbol{H}^{(0)}$ is the $n \times p$ feature matrix whose $i$-th row is $x_i$. For mini-batch gradient descent training, the training loss for an $L$-layer GCN is defined as $\frac{1}{|\mathcal{V}_{batch}|} \sum_{v_i \in \mathcal{V}_{batch}} \ell(y_i, z_i^{(L)})$, where $\ell$ is the loss function, batch nodes $\mathcal{V}_{batch}$ is a subset of $\mathcal{V}_{train}$ at each iteration. $z_i^{(L)}$ is the $i$-th row in $\boldsymbol{Z}^{(L)}$, $|\cdot|$ denotes the cardinality of a set.

In this paper, we set $\boldsymbol{P} = \tilde{\boldsymbol{D}}^{-1/2}(\boldsymbol{A} + \boldsymbol{I})\tilde{\boldsymbol{D}}^{-1/2}$, where $\tilde{\boldsymbol{D}}$ is a diagonal matrix with $\boldsymbol{D}_{ii} = 1 + \sum_i \boldsymbol{A}_{ij}$. The matrix $\boldsymbol{P}$ is constructed as a *renormalized Laplacian matrix* to help alleviate overfitting and exploding/vanishing gradients issues (Kipf & Welling, 2017), which is previously used by Kipf & Welling (2017); Chen et al. (2018a); Cong et al. (2020).

## 2.2 Layer-wise Sampling

To address the "neighbor explosion" issue for graph neural networks, sampling methods are integrated into the stochastic training. Motivated by the idea to approximate the matrix $\boldsymbol{P}\boldsymbol{H}^{(l)}$ in (1), FastGCN (Chen et al., 2018b) applies an importance-sampling-based strategy. Instead of individually sampling neighbors for each node in the $l$-th layer, they sample a set of $s$ neighbors $\mathcal{S}^{(l)}$ from $\mathcal{V}$ with importance sampling probability $p_i$, where $p_i \propto \sum_{j=1}^{n} \boldsymbol{P}_{ji}^2$ and $\sum_{i=1}^{n} p_i = 1$. For the $(l-1)$-th layer, they naturally set $\mathcal{V}^{(l-1)} = \mathcal{S}^{(l)}$. LADIES (Zou et al., 2019) improves the importance sampling probability $p_i$ as

$$p_i^{(l)} \propto \sum_{v_j \in \mathcal{N}^{(l)}} \boldsymbol{P}_{ji}^2, \forall i \in [n] \tag{2}$$

where $\mathcal{N}^{(l)} = \cup_{v_i \in \mathcal{V}^{(l)}} \mathcal{N}(v_i)$ and $\sum_{i=1}^{n} p_i^{(l)} = 1$. In this case, $\mathcal{S}^{(l)}$, the nodes sampled for the $l$-th layer, are guaranteed to be within the neighborhood of $\mathcal{V}^{(l)}$. The whole procedure can be concluded by introducing a diagonal matrix $\boldsymbol{S}^{(l)} \in \mathbb{R}^{n \times n}$ and a row selection matrix $\boldsymbol{Q}^{(l)} \in \mathbb{R}^{s_l \times n}$, which are defined as

$$\boldsymbol{Q}_{k,j}^{(l)} = \begin{cases} 1, & j = i_k^{(l)} \\ 0, & \text{else} \end{cases}, \quad \boldsymbol{S}_{j,j}^{(l)} = \begin{cases} (s_l p_{i_k^{(l)}}^{(l)})^{-1}, & j = i_k^{(l)} \\ 0, & \text{else,} \end{cases} \tag{3}$$

where $s_l$ is the sample size in the $l$-th layer and $\{i_k^{(l)}\}_{k=1}^{s_l}$ are the indices of rows selected in the $l$-th layer. The forward propagation with layer-wise sampling can thus be equivalently represented as $\tilde{\boldsymbol{Z}}^{(l+1)} = \boldsymbol{Q}^{(l+1)}\boldsymbol{P}\boldsymbol{S}^{(l)}\boldsymbol{H}^{(l)}\boldsymbol{W}^{(l)}, \boldsymbol{H}^{(l)} = (\boldsymbol{Q}^{(l)})^T\sigma(\tilde{\boldsymbol{Z}}^{(l)})$, where $\tilde{\boldsymbol{Z}}^{(l+1)}$ is the approximation of the embedding matrix for layer $l$.

# 3 Experimental Setup

History-oblivious layer-wise sampling methods, the subject of this work, rely on specific assumptions to construct the sampling probabilities. A paradox here is that the original LADIES (the model we aim to improve on) would automatically be optimal under its own assumptions. However, it is important to verify

their assumptions on common real-world open benchmarks, where our major empirical evaluation will be performed (c.f. Section 4). In advance of discussions on existing issues and corresponding remedies in Section 4 and Section 5, we introduce the basic setups of main experiments and datasets across the paper. Details about GCN model training are deferred to the related sections.

Table 1: Summary of datasets. Each undirected edge is counted once. Each node in ogbn-proteins has 112 binary labels. "Deg." refers to the average degree of the graph. "Feat" refers to the number of features. "Split Ratio" refers to the ratio of training/validation/test data.

| Dataset | Nodes | Edges | Deg. | Feat. | Classes | Tasks | Split Ratio | Metric |
|---|---|---|---|---|---|---|---|---|
| Reddit | 232,965 | 11,606,919 | 50 | 602 | 41 | 1 | 66/10/24 | F1-score |
| ogbn-arxiv | 160,343 | 1,166,243 | 13.7 | 128 | 40 | 1 | 54/18/28 | Accuracy |
| ogbn-proteins | 132,534 | 39,561,252 | 597.0 | 8 | binary | 112 | 65/16/19 | ROC-AUC |
| ognb-mag | 736,389 | 5,396,336 | 7.3 | 128 | 349 | 1 | 85/9/6 | Accuracy |
| ogbn-products | 2,449,029 | 61,859,140 | 50.5 | 100 | 47 | 1 | 8/2/90 | Accuracy |

**Benchmarks.** The distribution of the input graph will impact the effectiveness of sampling methods to a great extent—we can always construct a graph in an adversarial manner to favor one while deteriorate another. To overcome this issue, we conduct empirical experiments on 5 large real-world datasets to ensure the fair comparison and the representative results. The datasets (see details in Table 1) involve: Reddit (Hamilton et al., 2017), ogbn-arxiv, ogbn-proteins, ogbn-mag, and ogbn-products (Hu et al., 2020). Reddit is a traditional large graph dataset used by Chen et al. (2018b); Zou et al. (2019); Chen et al. (2018a); Cong et al. (2020); Zeng et al. (2020). Ogbn-arxiv, ogbn-proteins, ogbn-mag, and ogbn-products are proposed in Open Graph Benchmarks (OGB) by Hu et al. (2020). Compared to traditional datasets, the OGB datasets we use have a larger volume (up to the million-node scale) with a more challenging data split (Hu et al., 2020). The metrics in Table 1 follow the choices of recent works and the recommendation by Hu et al. (2020).

**Main experiments.** To study the influence of the aforementioned issues, we evaluate the matrix approximation error (c.f. Section 4.3 and Figure 2) of different methods in one-step propagation. This is an intuitive and useful metric to reflect the performance of the sampling strategy on approximating the original mini-batch training. Since the updates of parameters in the training are not involved in the simple metric above, in Section 6 we further evaluate the prediction accuracy on testing sets of both intermediate models during training and final outputs, using the metrics in Table 1.

## 4 Reconsider Importance Sampling Probabilities in Layer-wise Sampling

The efficiency of layer-wise sampling relies on its importance sampling procedure, which helps approximate node aggregations with much fewer nodes than involved. As expected, the choice of sampling probabilities can significantly impact the ultimate prediction accuracy of GCNs, and different sampling paradigms more or less seek to minimize the following variance (for the sake of notational brevity, from now on we omit the superscript $(l)$ when the objects are from the same layer)

$$\mathbb{E} \|\boldsymbol{QPSHW} - \boldsymbol{QPHW}\|_F^2, \tag{4}$$

where $\|\cdot\|_F$ denotes the Frobenius norm. Under the layer-dependent sampling framework, Zou et al. (2019) show that the optimal sampling probability $p_i$ for node $i$ satisfies (see Appendix C.1 for a derivation from a perspective of approximate matrix multiplication)

$$p_i \propto \|\boldsymbol{QP}^{[i]}\| \cdot \|(\boldsymbol{HW})_{[i]}\|, \tag{5}$$

where for a matrix $\boldsymbol{A}$, $\boldsymbol{A}_{[i]}$ and $\boldsymbol{A}^{[i]}$ respectively represent the $i$-th row/column of matrix $\boldsymbol{A}$.

### 4.1 Current Strategies and Their Limitation

The optimal sampling probabilities (5) discussed above are usually unavailable during the mini-batch gradient descent training due to a circular dependency: to sample the nodes in the $\ell$-th layer based the probabilities

in Equation (5), we need the hidden embedding $\boldsymbol{H}^{(\ell)}$ which in turn depends on the nodes not yet sampled in the $(\ell-1)$-th layer. In this case, FastGCN (Chen et al., 2018b) and LADIES (Zou et al., 2019) choose to perform layer-wise importance sampling without the information from $\boldsymbol{HW}$ [1]. In particular, FastGCN (resp. LADIES) assumes $\|(\boldsymbol{HW})_{[i]}\| \propto \|\boldsymbol{P}^{[i]}\|$ (resp. $\|\boldsymbol{QP}^{[i]}\|$), and sets their sampling probabilities as $p_i \propto \|\boldsymbol{P}^{[i]}\|^2$ (resp. $\|\boldsymbol{QP}^{[i]}\|^2$), $\forall i \in [n]$.

The proportionality assumption above seems sensible considering the computation of the hidden embedding $\boldsymbol{H}$ involves $\boldsymbol{P}$. However, this assumption is unguaranteed because of the changing weight matrix $\boldsymbol{W}$ in training, and no previous work (to our knowledge) scrutinizes whether this assumption generally holds. To study the appropriateness of the core assumption, we conduct a linear regression [2]

$$\boldsymbol{y} \sim \beta_0 + \beta_1 \boldsymbol{x} \tag{6}$$

for each layer separately, where $x$ ranging over $\|(\boldsymbol{HW})_{[i]}\|$'s is the $\ell^2$ norm of a certain row in $\boldsymbol{HW}$ and $y$ over $\|\boldsymbol{P}^{[i]}\|$ is the norm of the corresponding column in $\boldsymbol{P}$.

Table 2: Regression coefficients in Equation (6) for 3-layer GCNs trained with LADIES and full-batch SGD respectively. Negative $\beta_1$'s are highlighted in boldface.

| Method | | LADIES | | | | Full-batch | | | |
|---|---|---|---|---|---|---|---|---|---|
| Dataset | | ogbn-arxiv | reddit | ogbn-proteins | ogbn-mag | ogbn-arxiv | reddit | ogbn-proteins | ogbn-mag |
| | $\beta_0$ | $3.517 \pm 0.002$ | $11.34 \pm 0.01$ | $4.162 \pm 0.001$ | $3.687 \pm 0.001$ | $2.364 \pm 0.002$ | $29.53 \pm 0.03$ | $3.942 \pm 0.001$ | $3.282 \pm 0.001$ |
| Layer 1 | $\beta_1$ | **-0.54 $\pm$ 0.01** | $8.03 \pm 0.07$ | $0.488 \pm 0.004$ | **-0.391 $\pm$ 0.004** | **-0.15 $\pm$ 0.01** | $15.66 \pm 0.17$ | $0.375 \pm 0.004$ | **-1.03 $\pm$ 0.03** |
| | $R^2$ | $0.012$ | $0.012$ | $0.023$ | $0.003$ | $0.001$ | $0.008$ | $0.013$ | $0.026$ |
| | $\beta_0$ | $6.21 \pm 0.01$ | $4.41 \pm 0.01$ | $26.95 \pm 0.01$ | $10.67 \pm 0.01$ | $4.01 \pm 0.01$ | $27.94 \pm 0.02$ | $23.46 \pm 0.01$ | $10.26 \pm 0.01$ |
| Layer 2 | $\beta_1$ | $4.20 \pm 0.03$ | $4.59 \pm 0.05$ | **-38.18 $\pm$ 0.17** | $1.58 \pm 0.03$ | $4.47 \pm 0.02$ | $24.07 \pm 0.02$ | **-35.09 $\pm$ 0.15** | **-4.12 $\pm$ 0.01** |
| | $R^2$ | $0.028$ | $0.008$ | $0.074$ | $0.001$ | $0.051$ | $0.022$ | $0.081$ | $0.024$ |
| | $\beta_0$ | $22.21 \pm 0.02$ | $4.72 \pm 0.01$ | $104.924 \pm 0.03$ | $29.86 \pm 0.03$ | $19.72 \pm 0.02$ | $45.82 \pm 0.03$ | $174.72 \pm 0.09$ | $41.98 \pm 0.02$ |
| Layer 3 | $\beta_1$ | $1.00 \pm 0.06$ | $0.10 \pm 0.03$ | **-137.8 $\pm$ 0.4** | **-0.13 $\pm$ 0.08** | $2.16 \pm 0.07$ | $9.49 \pm 0.19$ | **-367.2 $\pm$ 1.1** | **-29.14 $\pm$ 0.05** |
| | $R^2$ | $< 0.001$ | $< 0.001$ | $0.160$ | $< 0.001$ | $0.001$ | $0.002$ | $0.153$ | $0.100$ |

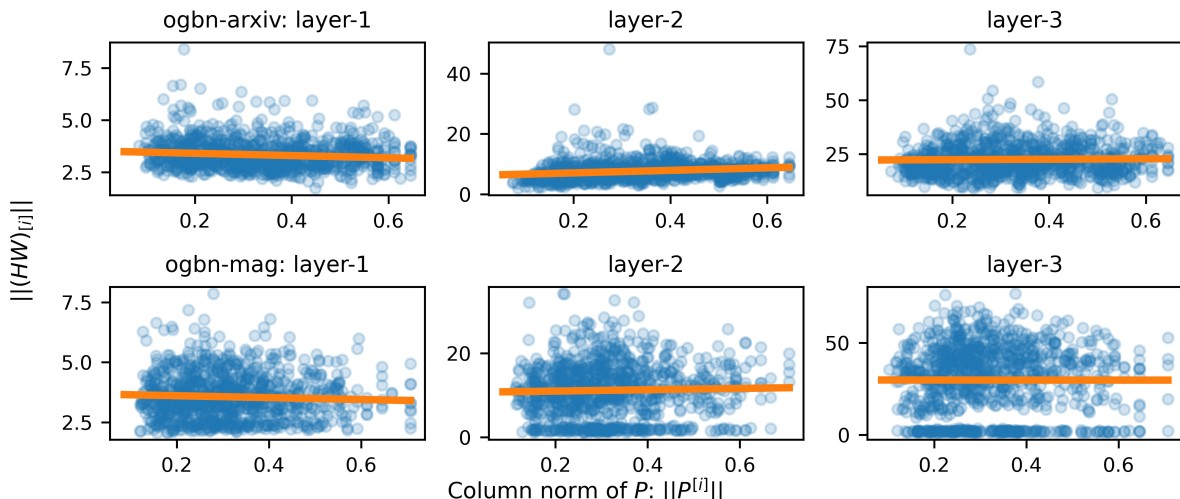

Figure 1: Regression lines (orange) and scatter plots of $\|(\boldsymbol{HW})_{[i]}\|$ in 3-layer LADIES's different layers on ogbn-arxiv and ogbn-mag.

---

[1]This scheme decouples the sampling and the training procedure. We can save the training runtime on GPU by preparing sampling on CPU in advance.

[2]We take the counterpart assumption $\|(\boldsymbol{HW})_{[i]}\| \propto \|\boldsymbol{QP}^{[i]}\|$ in LADIES as a randomized version of the one $\|(\boldsymbol{HW})_{[i]}\| \propto \|\boldsymbol{P}^{[i]}\|$ in FastGCN (and therefore focus on the latter for the regression experiments), since $\|(\boldsymbol{HW})_{[i]}\|$'s are conceptually independent of the subsequent row selection matrix $\boldsymbol{Q}$. Supplementary regression experiments on $\|(\boldsymbol{HW})_{[i]}\| \|\boldsymbol{QP}^{[i]}\|$ are collected in Appendix D.

Figure 1 presents regression curves of each layer in LADIES on ogbn-arxiv and ogbn-mag. Table 2 summarizes the regression coefficient $\beta_0$, $\beta_1$, and $R^2$ of a 3-layer GCN trained by LADIES or regular mini-batch SGD without node sampling ("full-batch" in short). The regression results demonstrate that the assumption $\|(\boldsymbol{HW})_{[i]}\| \propto \|\boldsymbol{P}^{[i]}\|$ is violated on many real-world datasets from two-fold evidence: negative regression coefficients and small $R^2$'s. First, regression coefficients for full-batch SGD and LADIES show similar patterns: negative slope $\beta_1$'s appear across multiple layers and different datasets, such as layer 1 of ogbn-arxiv, layer 2 of ogbn-protains, and both layers 1 and 3 of ogbn-mag. The negative correlation clearly violates the assumption, $\|(\boldsymbol{HW})_{[i]}\| \propto \|\boldsymbol{P}^{[i]}\|$. Secondly, the $R^2$ (see Table 2) in the single variable regression is $corr^2(x, y)$, which measures the proportion of $y$'s variance explained by $x$. A positive $\beta_1$ with small $R^2$ can only imply a positive but weak correlation between $\|(\boldsymbol{HW})_{[i]}\|$ and $\|\boldsymbol{P}^{[i]}\|$, however, a weak correlation cannot imply a proportionality relationship. For example, even though the $\beta_1$ is positive for each layer in Reddit data, corresponding $R^2$'s are at the scale of 0.01. Consequently, the performance of LADIES on Reddit is not surprisingly inferior to our method with new sampling probabilities (LADIES+flat in Table 4). Experimental setups are collected in Appendix A.4. Additional regression results for FastGCN and our proposed methods are collected in Appendix D.

## 4.2 Proposed Sampling Probabilities

The small or even negative correlation between $\|(\mathbf{HW})_{[i]}\|$ and $\|\mathbf{P}^{[i]}\|$ ($\|\mathbf{QP}^{[i]}\|$) implies the inappropriateness of the proportionality assumption and the resulting sampling probabilities in FastGCN/LADIES. To address this issue, we instead admit that we have limited prior knowledge of $\boldsymbol{HW}$ under the history-oblivious setting, and follow the Principle of Maximum Entropy to assume a uniform distribution of $\|(\boldsymbol{HW})_{[i]}\|$'s. With this belief, we propose the following sampling probabilities:

$$p_i \propto \|\boldsymbol{QP}^{[i]}\|, \quad \forall i \in [n]. \tag{7}$$

Compared to LADIES in Equation (2), our proposed sampling probabilities $p_i$'s are more conservative. From a matrix approximation perspective, we rewrite the target matrix product as $\boldsymbol{QPIHW}$, and only aim to approximate the known part $\boldsymbol{QPI}$. It turns out that assuming the uniform distribution of the norms of rows in $\boldsymbol{HW}$ can help improve both the variance of the matrix approximation and the prediction accuracy of GCNs, as empirically shown in Section 4.3 and Section 6.

In addition to the empirical results, we compare the estimation variance of our probabilities with LADIES in Lemma C.1 under a mild condition [3]. Specifically, a common long-tail distribution of $\boldsymbol{HW}$ can justify the strengths of our new probabilities. More discussion and visualization on the distribution of $\boldsymbol{HW}$ are provided in Appendix C.

## 4.3 Matrix Approximation Error Evaluation

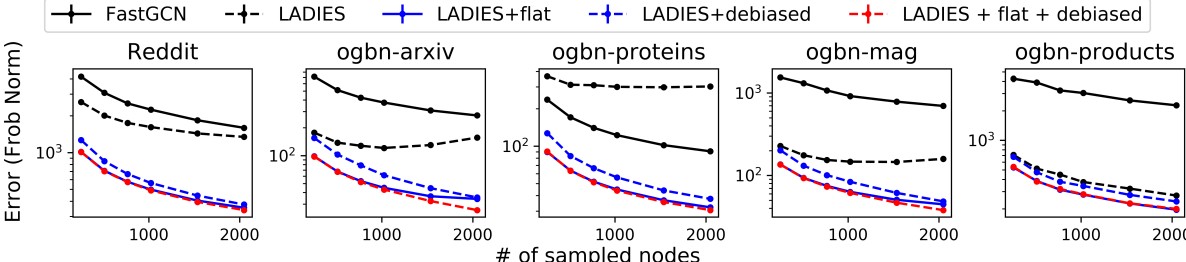

Figure 2: Matrix approximation errors of layer-wise sampling methods. The error curves of LADIES show an abnormal U-shape on ogbn-arxiv and ogbn-mag datasets. "flat" and "debiased" denote our proposed methods in Sections 4 and 5 respectively.

---

[3]We reiterate that the variance depends on the underlying distribution of the row norms of $\boldsymbol{HW}$, and therefore no sampling probabilities can always have smaller variance than others.

To further justify our proposed sampling probabilities, we consider the following 1-layer embedding approximation error, which evaluates the propagation approximation to the embedding aggregation of a batch:

$$\|\tilde{\boldsymbol{Z}}_{batch}^{(1)} - \tilde{\boldsymbol{Z}}_{sampling}^{(1)}\|_F = \|\boldsymbol{Q}_{batch}\boldsymbol{P}\boldsymbol{H}^{(0)}\boldsymbol{W}^{(0)} - \boldsymbol{Q}_{batch}\boldsymbol{P}\boldsymbol{S}\boldsymbol{H}^{(0)}\boldsymbol{W}^{(0)}\|_F,$$

where $\tilde{\boldsymbol{Z}}_{batch}^{(1)}$ and $\tilde{\boldsymbol{Z}}_{sampling}^{(1)}$ are the embedding at the first layer computed using all available neighbors and a certain sampling method, respectively; $\boldsymbol{S}$ is the sampling matrix; $\boldsymbol{Q}_{batch}$'s $0, 1$ diagonal entries indicate if a node is in the batch. The experiments are repeated 200 times, in which we regenerate the batch nodes (shared by all sampling methods) and the sampling matrix for each method. The batch size is fixed as 512, and the numbers of sampled neighbors ranges over $256, 512, 768, 1024, 1536,$ and $2048$. $\boldsymbol{W}^{(0)}$ is fixed and inherited from the trained model reported in Section 6.

In Figure 2, the result of our proposed sampling probabilities (denoted as "LAIDES+flat", blue solid line) is consistently better than that of the original LADIES method (black dashed line) and of FastGCN (black solid line) on every dataset. The debiasing method in this figure will be discussed shortly in the next section.

## 5 Debiased Sampling

We first make the clarification that in the derivation of previous layer-wise sampling strategies the neighbor nodes are sampled *with* replacement, proven to be unbiased approximations of GCN embedding. However, in their actual implementations, sampling is always performed *without* replacement, which induces biases since the estimator remain the same as sampling with replacement. We illustrate the biases in Figure 2, where the matrix approximation errors on ogbn-arxiv and ogbn-mag datasets (sparse graphs with few average degrees) are U-shape for LADIES. The curve indicates that the errors even increase with the number of sub-samples, and the approximation performance is heavily deteriorated by the biases.

In the following subsections, we dive into the implementation of FastGCN and LADIES, reveal the origin of the biases, propose a new debiasing method for sampling without replacement, and study the statistical properties of the debiased estimators.

**Remark.** We insist on sampling without replacement because it helps reduce the variance of the estimator. For instance, current node-wise sampling GCNs also sample "without replacement"—they apply simple random sampling (SRS, sampling with all-equal probabilities), which is guaranteed to shrink the variance by a finite population correction (FPC) factor $\frac{n-s}{n-1}$ (Lohr, 2019, Section 2.3) [4].

### 5.1 Weighted Random Sampling (WRS)

The implementation of layer-wise importance sampling (without replacement) follows a sequential procedure named as weighted random sampling (WRS) (Efraimidis & Spirakis, 2006, Algorithm D). Given a set $V = [n]$ representing the indices of $n$ items $\{\boldsymbol{X}_i\}_{i=1}^n$ [5] and the associated sampling probabilities $\{p_i\}_{i=1}^n$, we sample $s$ samples and denote the sampled indices as $I_k$ for $k = 1, 2, ..., s$. With $s$ sampled indices, FastGCN/LADIES use the following importance sampling estimator to approximate the target sum of matrices $\sum_{i=1}^n \boldsymbol{X}_i$ (adapted to the notations in this section)

$$\frac{1}{s}\sum_{k=1}^s \boldsymbol{X}_{I_k}/p_{I_k}, \tag{8}$$

which is a weighted average of $\boldsymbol{X}_{I_k}$'s and induces biases when obtained through sampling without replacement. We aim to preserve the linear form $\boldsymbol{Y}_s := \sum_{k=1}^s \beta_k \boldsymbol{X}_{I_k}$ in debiasing, and develop new coefficients $\beta_k$ for each $\boldsymbol{X}_{I_k}$ to make $Y_s$ unbiased; the debiasing algorithm is officially presented in Section 5.3.

---

[4] $s$ and $n$ denote the sample size and the population size.

[5] In layer-wise sampling, $\boldsymbol{X}_i$ represents $\boldsymbol{B}^{[i]}\boldsymbol{C}_{[i]}^T$, where for brevity $\boldsymbol{QP}$ ($\boldsymbol{HW}$) is denoted as $\boldsymbol{B}$ ($\boldsymbol{C}$) throughout the analysis.

## 5.2 Analysis of Bias

To analyze the bias of Equation (8), we introduce the following auxiliary notations. In WRS, given the set $S_k$ of $k$ previously sampled indices ($0 \leq k \leq s - 1, S_0 := \emptyset$), the $(k+1)$-th random index $I_{k+1}$ is sampled from the set $V - S_k$ of the rest $n - k$ indices with probabilities

$$p_i^{(0)} := p_i, \forall i \in V = [n]; \quad p_i^{(k)} := \frac{p_i}{\sum_{j \in V - S_k} p_j}, \forall k \in [s-1], i \in V - S_k.$$

With the notations introduced, we are now able to analyze the effect of applying Equation (8) while the WRS algorithm is performed. The expectation of a certain summand $\boldsymbol{X}_{I_{k+1}}/p_{I_{k+1}}$ will be

$$\mathbb{E} \frac{\boldsymbol{X}_{I_{k+1}}}{p_{I_{k+1}}} = \mathbb{E} \left[ \mathbb{E} \left[ \frac{\boldsymbol{X}_{I_{k+1}}}{p_{I_{k+1}}^{(k)}} \frac{p_{I_{k+1}}^{(k)}}{p_{I_{k+1}}} \mid \mathcal{F}_k \right] \right] = \mathbb{E} \left[ \frac{1}{\sum_{i \in V - S_k} p_i} \sum_{i \in V - S_k} \boldsymbol{X}_i \right], \tag{9}$$

where $\mathcal{F}_k$ is the $\sigma$-algebra generated by the random indices inside the corresponding set $S_k, \forall k = 0, 1, \cdots, s - 1$, and the second equation holds because $\frac{p_{I_{k+1}}^{(k)}}{p_{I_{k+1}}} = \frac{1}{\sum_{i \in V - S_k} p_i}$ is $\mathcal{F}_k$-measurable. The expectation is in general unequal to the target $\sum_{i=1}^n \boldsymbol{X}_i$ for $k > 0$, except for some extreme conditions, such as all-equal $p_i$'s. The bias in each summand (except for the first term with $k = 0$) accumulates and results in the biased estimation.

## 5.3 Debiasing Algorithms

We start with a review of existing works on debiasing algorithms for stochastic gradient estimators. Given a sequence of random indices sampled through WRS, there are two common genres to assign coefficients to summands in Equation (8). Both of the two genres relate to the *stochastic sum-and-sample* estimator (Liang et al., 2018; Liu et al., 2019), which can be derived from Equation (9). Using the fact $\mathbb{E} \frac{\boldsymbol{X}_{I_{k+1}}}{p_{I_{k+1}}} \sum_{i \in V - S_k} p_i = \mathbb{E} \left[ \sum_{i \in V - S_k} \boldsymbol{X}_i \right]$, a stochastic sum-and-sample estimator of $\sum_{i=1}^n \boldsymbol{X}_i$ can be immediately constructed as [6]

$$\boldsymbol{\Pi}_{k+1} = \sum_{j \in S_k} \boldsymbol{X}_i + \frac{\boldsymbol{X}_{I_{k+1}}}{p_{I_{k+1}}^{(k)}}, \forall k = 0, 1, \cdots s - 1. \tag{10}$$

To minimize the variance, Liang et al. (2018); Liu et al. (2019) develop the first genre to focus on the last estimator $\boldsymbol{\Pi}_s$ and propose methods to pick the initial $s - 1$ random indices. Kool et al. (2020, Theorem 4) turn to the second genre which utilize Rao-Blackwellization (Casella & Robert, 1996) of $\boldsymbol{\Pi}_s$.

In fast training for GCN, both of the two genres are somewhat inefficient from a practitioner's perspective. The first genre works well when $\sum_{i \in S_{s-1}} p_i$ is close to 1, otherwise the last term in $\boldsymbol{\Pi}_s$, $\frac{\boldsymbol{X}_{I_{k+1}}}{p_{I_{k+1}}^{(k)}}$, will bring in large variance and reduce the sample efficiency; for the second genre, the time cost to perform Rao-Blackwellization (Kool et al., 2020) is extremely high ($\mathcal{O}(2^s)$ even with approximation by numerical integration) and conflicts with the purpose of fast training. To overcome the issues of the two existing genres, we propose an iterative method to fully utilize each estimator $\boldsymbol{\Pi}_{k+1}$ with acceptable runtime to decide the coefficients for each term in Equation (8).

Denote our final estimator with $s$ samples as $\boldsymbol{Y}_s$. Algorithm 1 returns the coefficients $\beta_k$'s used in the debiased estimator $\boldsymbol{Y}_s = \sum_{k=1}^s \beta_k \boldsymbol{X}_{I_k}$. The main idea is to perform recursive estimation $\boldsymbol{Y}_1, \boldsymbol{Y}_2, ...$ until $\boldsymbol{Y}_s$ and thus update $\beta$ accordingly. To be more specific, we recursively perform the weighted averaging below:

$$\boldsymbol{Y}_0 := 0, \quad \boldsymbol{Y}_{k+1} := (1 - \alpha_{k+1})\boldsymbol{Y}_k + \alpha_{k+1}\boldsymbol{\Pi}_{k+1}, \forall k = 0, 1, \cdots, s - 1,$$

where $\alpha_1 = 1$ and $\alpha_{k+1}$ is a constant depending on $k$. Specifically, $\boldsymbol{Y}_1 = \boldsymbol{\Pi}_1 = \boldsymbol{X}_{I_1}/p_{I_1}$ is unbiased and the unbiasedness of $\boldsymbol{Y}_s$ can be obtained by induction as each $\boldsymbol{\Pi}_{k+1}$ is unbiased as well. There can be variant choices of $\alpha_{k+1}$'s. For example, the *stochastic sum-and-sample* estimator (10) sets all $\alpha_{k+1}$'s as 0 except for $\alpha_s = 1$. In contrast, we intentionally specify $\alpha_{k+1} = \frac{n}{(n-k)(k+1)}$, motivated by the preference that if all $p_i$'s are $1/n$, the output coefficients of the algorithm will be all $1/s$, the same as the ones in an SRS setting.

---

[6]The proof of its unbiasedness is brief and provided by Kool et al. (2020, Appendix C.1).

---

**Algorithm 1:** Iterative updates of coefficients to construct the ultimate debiased estimator $\boldsymbol{Y}_s$.

---

**Input:** probabilities $\{p_i\}_{i=1}^n$, random indices $\{I_{k+1}\}_{k=0}^{s-1}$ generated by WRS with $\{p_i\}_{i=1}^n$

**Output:** a length $s$ coefficient vector $\boldsymbol{\beta}$

Initialize $\boldsymbol{\beta} = \boldsymbol{0} \in \mathbb{R}^s$, $p_S = 0$ (sum of probabilities);

**for** $k \leftarrow 0$ **to** $s-1$ **do**

$\quad \alpha_{k+1} = \frac{n}{(n-k)(k+1)}$;

$\quad \boldsymbol{\beta}_{[k+1]} = \alpha_{k+1}(1-p_S)/p_{I_{k+1}}$;

$\quad$ **for** $j \leftarrow 0$ **to** $k-1$ **do**

$\quad\quad \boldsymbol{\beta}_{[j+1]} = (1-\alpha_{k+1})\boldsymbol{\beta}_{[j+1]} + \alpha_{k+1}$;

$\quad$ **end**

$\quad p_S = p_S + p_{I_{k+1}}$;

**end**

return $\boldsymbol{\beta}$;

---

## 5.4 Effects of Debiasing

We evaluate the debiasing algorithm again by matrix approximation error (see Section 4.3). As shown in Figure 2, our proposed debiasing method can significantly improve the one-step matrix approximation error on all datasets. In particular, by introducing the debiasing algorithm, the U-shape curve of LADIES in Figure 2 no longer exists for debiased LADIES.

We also observe if the new sampling probabilities have already been applied (LADIES + flat in Figure 2), an additional debiasing algorithm (LADIES + flat + debiased) only makes marginal improvement on sparse graphs (ogbn-arxiv and ogbn-mag). The observation implies that the effect of debiasing and new sampling probabilities may have overlaps.

We provide the following conjectures for this phenomenon. First, for the bias introduced by sampling without replacement, it is significant only when the proportion of sampled nodes over all neighbor nodes is large enough. With a fixed batch size while an increasing sample size, sparse graphs generally exhibit a larger bias since they have a larger "sampling proportion" than dense graphs. Second, our proposed sampling probabilities have a flatter distribution than LADIES, which resembles a uniform distribution and implies a smaller bias (there is no bias in SRS). The phenomenon, therefore, implies an efficient practice to apply the debiasing algorithm: users can decide whether to debias the estimation based on the ratio of the sampling size to the batch size and the degrees in the graphs.

In addition to the effect on matrix approximation, the evidence that the debiasing algorithm can also accelerate the convergence and improve the model prediction accuracy will be provided in Section 6.

## 5.5 Sampling Time

The iterative updates in Algorithm 1 induces additional $\mathcal{O}(s^2)$ time complexity cost in sampling per batch, as in the $k$-th iteration we need to update the coefficients for the first $k$ random indices sampled. We first remark the time complexity is comparable to the one of embedding aggregation in layer-wise training, as shown in Appendix B. Moreover, since our layer-wise sampling procedure can be performed independently on CPU, it will not retard the training on GPU [7]. Note that this decoupling of sampling and training does not hold for some node-wise or layer-wise sampling methods, such as VR-GCN, which requires up-to-date embedding information.

The experimental results for sampling time in Table 3 further show that the additional cost of debiasing is acceptable compared to FastGCN and LADIES. For example, comparing "LADIES + debiasing" to LADIES, the sampling time only increases from 11.2 ms to 13.6 ms on ogbn-proteins. In contrast, vanilla node-wise sampling takes 831 ms due to the overhead of row-wisely sampling the sparse Laplacian matrix.

---

[7]Technically the sampling results can be prepared in advance of the training on the GPU, and therefore we claim sampling can be performed independently of training. Furthermore, we remark the sequential debiasing procedure is not a fit for GPU.

Table 3: Average sampling time (in milliseconds) per batch for layer-wise methods and the vanilla node-wise method. The experiment is conducted on CPU. The batch size is set as 512 and the sample size is set as 512/1024 (indicated in the following parentheses). The "f" and "d" in "LADIES+f+d" denotes "flat" and "debiased" respectively. More experimental details are collected in Appendix A.3.

|  | FastGCN | LADIES | LADIES+f | LADIES+d | LADIES+f+d | Node-wise |
|---|---|---|---|---|---|---|
| Reddit (512) | $10.6 \pm 0.9$ | $10.9 \pm 0.3$ | $10.0 \pm 0.2$ | $13.1 \pm 0.3$ | $13.1 \pm 0.4$ | $632.5 \pm 4.3$ |
| Reddit (1024) | $10.0 \pm 0.6$ | $11.8 \pm 0.4$ | $10.4 \pm 0.1$ | $17.1 \pm 0.4$ | $16.1 \pm 0.6$ | $637.0 \pm 4.2$ |
| arxiv (512) | $4.2 \pm 0.1$ | $8.3 \pm 0.1$ | $7.8 \pm 0.1$ | $11.7 \pm 0.2$ | $11.7 \pm 0.5$ | $585.2 \pm 3.6$ |
| arxiv (1024) | $6.9 \pm 0.1$ | $9.7 \pm 0.1$ | $9.0 \pm 0.1$ | $17.2 \pm 0.3$ | $16.5 \pm 0.3$ | $585.6 \pm 3.1$ |
| mag (512) | $16.8 \pm 0.1$ | $27 \pm 0.1$ | $24.5 \pm 0.1$ | $30.0 \pm 0.03$ | $27.8 \pm 0.1$ | $1084.3 \pm 1.7$ |
| mag (1024) | $18.9 \pm 0.2$ | $28.6 \pm 0.2$ | $27.7 \pm 0.1$ | $36.0 \pm 0.2$ | $34.9 \pm 0.1$ | $1119 \pm 2.9$ |
| proteins (512) | $11.1 \pm 1$ | $11.2 \pm 0.3$ | $10 \pm 0.2$ | $13.6 \pm 0.2$ | $12.6 \pm 0.2$ | $830.9 \pm 5.3$ |
| proteins (1024) | $8.9 \pm 0.2$ | $12.4 \pm 0.1$ | $11.4 \pm 0.1$ | $18.9 \pm 0.2$ | $18.0 \pm 0.3$ | $804.2 \pm 4.6$ |
| products (512) | $54.8 \pm 0.7$ | $83.4 \pm 1.3$ | $80.3 \pm 0.4$ | $83.7 \pm 0.8$ | $83.5 \pm 0.6$ | $2795.4 \pm 4.7$ |
| products (1024) | $57.1 \pm 0.5$ | $80.8 \pm 0.8$ | $78.7 \pm 0.7$ | $87.0 \pm 0.6$ | $85.4 \pm 0.7$ | $2737.7 \pm 4.8$ |

## 5.6 Analysis of Variance

In sampling without replacement, the selected samples are no longer independent, and therefore the classical analysis in previous works (c.f. Lemma C.2 in Appendix C) cannot be applied to the variance of WRS-based estimators. To quantify the variance under the WRS setting, we leverage a common technique in experimental design—viewing $\{\beta_i^{(k)}\}_{i=1}^n, \forall k \in [s]$ as random variables. $\beta_i^{(k)}$ denote the coefficients assigned to $\boldsymbol{X}_i$'s when the $k$-th sample is drawn (if $i \notin S_k$, $\beta_i^{(k)} := 0$). This technique can derive the same result as in the previous random indices ($I_k$'s) setting, while allow a finer analysis of the variance. We can rewrite the variance in Equation (4) as [8]

$$\mathbb{E} \|\boldsymbol{BSC} - \boldsymbol{BC}\|_F^2 = \sum_{j,k} \operatorname{Var} \left( \sum_{i=1}^n \beta_i^{(s)} \boldsymbol{B}_j^{[i]} \boldsymbol{C}_{[i],k} \right).$$

The variance above is determined by the covariance matrix $\operatorname{Cov}(\boldsymbol{\beta})$, whose $(i,j)$-th element is $\operatorname{Cov}(\beta_i^{(s)}, \beta_j^{(s)})$. We provide the following proposition for the covariance matrix $\operatorname{Cov}(\boldsymbol{\beta})$ in Algorithm 1:

**Theorem 1.** *For all $k \in [s]$, let $p_{S_k}$ be the probability of having $S_k$ as the first $k$ samples, $\bar{q}_i^{(k)}$ be the probability of index $i$ not in the $k$ samples, and $\bar{q}_{i,j}^{(k)}$ be the probability of both index $i,j$ not in the $k$ samples. Define $r_i^{(k)} := \sum_{S_k \not\ni i} p_{S_k} (1 - \sum_{j \in S_k} p_j)$, where $\sum_{S_k \not\ni i}$ iterates over all $S_k$ that does not contain $i$. Then $\operatorname{Var}(\beta_i^{(k+1)}) \geq 0$ and $\operatorname{Cov}(\beta_i^{(k+1)}, \beta_j^{(k+1)}) \leq 0$ are recursively given as:*

$$(1 - \alpha_{k+1})^2 \operatorname{Var}(\beta_i^{(k)}) + \left( \frac{r_i^{(k)}}{p_i} - \alpha_{k+1}^2 \bar{q}_i^{(k)} \right), \tag{11}$$

$$(1 - \alpha_{k+1})^2 \operatorname{Cov}(\beta_i^{(k)}, \beta_j^{(k)}) - \alpha_{k+1}^2 \bar{q}_{i,j}^{(k)}. \tag{12}$$

*Furthermore, there exists a sequence $\{\alpha_k\}_{k=1}^s$ only depending on $k, n$ such that for all $i, j$, $Var(\beta_i^{(k)}) \leq \frac{1}{k}(\frac{1}{p_i} - 1), |Cov(\beta_i^{(k)}, \beta_j^{(k)})| \leq \frac{1}{k}, \forall k \in [s]$.*

Proof and discussions are collected in Appendix C.4. We remark that due to the fixed weights $\alpha_s = 1$ in the stochastic sum-and-sample estimator (10), its (co)variance is usually larger than ours, especially when $s \ll n$ (intuitively the second term $\boldsymbol{X}_{I_s}/p_{I_s}^{(s-1)}$ in Equation (10) will cause large variance).

---

[8] We let $\boldsymbol{B}$ ($\boldsymbol{C}$) have $n$ columns (rows), and the $j(k)$-th element in the $i$-th column (row) is denoted as $\boldsymbol{B}_j^{[i]}$ ($\boldsymbol{C}_{[i],k}$).

# 6 Experiments

In this section, we empirically evaluate the performance of each method on five node prediction datasets: Reddit, ogbn-arxiv, ogbn-proteins, ogbn-mag, and ogbn-products (c.f. Table 1). We denote "LADIES+flat", "LADIES+debiased", and "LADIES+flat+debiased" respectively as the variants of LADIES with the improvements from Section 4, Section 5, and from both. We compare our methods to the original GCN with mini-batch stochastic training (denoted by full-batch), two layer-wise sampling methods: FastGCN and LADIES. Apart from that, we also implement several other fast GCN training methods, including Graph-SAGE (Hamilton et al., 2017) (vanilla node-wise sampling while keep using the GCN architecture), VR-GCN (Chen et al., 2018a), and a subgraph sampling method GraphSAINT (Zeng et al., 2020).

In training, we use a 2-layer GCN for each task trained with an ADAM optimizer. (Due to limited computational resources, we have to use the shallow GCN since the full-batch method and node-wise sampling methods require much more GPU memory even when $L = 3$.) The number of hidden variables is 256 and the batch size is 512. For layer-wise sampling methods, we consider two settings for node sample size:

1. fixed as 512 (equal to the batch size);

2. an "increasing" setting (denoted with a suffix (2)) that double nodes will be sampled in the next layer.

For node-wise sampling methods (GraphSAGE, VR-GCN), the sample size per node is 2 (denoted with a suffix (2)). For the subgraph sampling method GraphSAINT, the subgraph size is by default equal to the batch size. The experimental results are reported in Table 4, in the form of "mean(±std.)", computed based on 5 runs. More details of the settings are deferred to Appendix A.1.

## 6.1 Model Convergence Trajectory

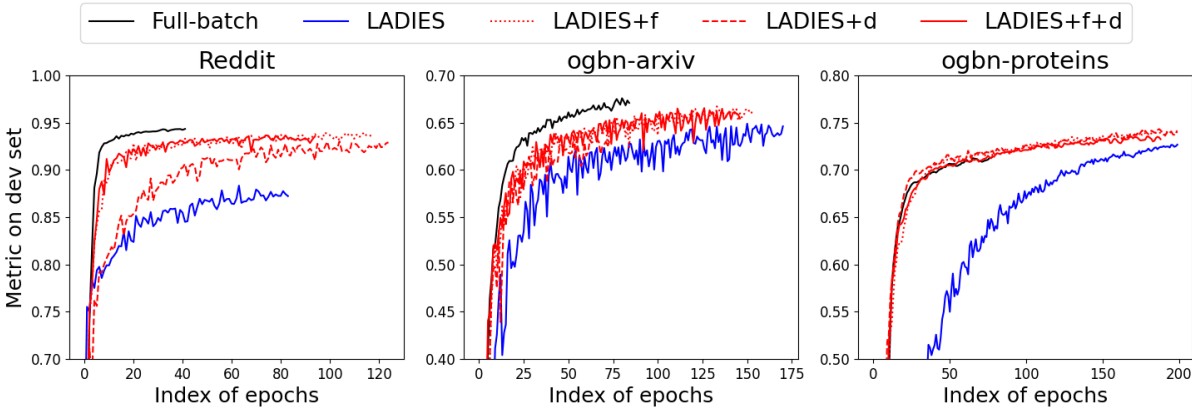

Figure 3: Metrics (detailed in Table 1) on the validation (dev) sets in each epoch. "f" and "d" refer to the "flat" sampling probability and "debiasing" respectively. The layer-wise sampling methods follow the "increasing" setting (denoted with a suffix (2) in Table 4).

We first compare the convergence rates of layer-wise methods. The convergence curves on ogbn-proteins and ogbn-products are shown in Figure 3 (FastGCN is excluded for clearer illustration, due to its outlying curve). Complete results of all methods (including node-wise and subgraph) on all five datasets are deferred to Figure 4 in Appendix A.2.

As shown in Figure 3 (and Figure 4 in the appendix), our proposed improvements (LADIES + flat, LADIES + debias, LADIES + flat + debias) exhibit faster convergence rate than LADIES (the solid blue curve). The observation implies both the new sampling probabilities ("flat") and the debiasing algorithm can help accelerate the convergence. Specifically, we note that the effect of debiasing is not as significant as choosing a proper sampling scheme on some datasets (e.g. Reddit and ogbn-products).

## 6.2 Prediction Accuracy

Table 4: Metrics (detailed in Table 1) on testing sets of benchmarks. The best results among layer-wise sampling methods and all methods are both highlighted in boldface. The metrics are in percentage (%). The averaged training time for one epoch is in milliseconds and measured on the GPU.

| | Metrics | | | | | Epoch avg. train. time | |
| | Reddit | ogbn-arxiv | ogbn-mag | ogbn-proteins | ogbn-products | ogbn-arxiv | ogbn-products |
|---|---|---|---|---|---|---|---|
| Full-batch | 93.81±0.18 | 66.39±0.25 | 29.60±0.27 | 65.71±0.11 | 68.33±0.16 | 65.2 ± 3.97 | 703 ± 77.8 |
| Node-wise (2) | 92.13±0.27 | 64.51±0.30 | 29.05±0.45 | 65.76±0.18 | 68.71±0.07 | 22.0 ± 3.64 | 8.34 ± 1.21 |
| VR-GCN (2) | **94.62±0.04** | **67.49±0.25** | 28.99±0.40 | 67.45±0.02 | **70.90±0.28** | 86.8 ± 6.09 | 88.5 ± 2.51 |
| GraphSAINT | 89.47±0.83 | 60.58±0.62 | 24.77±0.88 | 66.33±0.07 | 62.77±1.04 | 23.1 ± 3.26 | 8.26 ± 1.11 |
| FastGCN | 44.46±2.30 | 25.44±0.82 | 7.13±0.48 | 52.44±1.88 | 26.98±0.42 | 24.1 ± 5.12 | 7.43 ± 1.04 |
| LADIES | 73.86±0.17 | 60.95±0.31 | 24.79±0.48 | 68.28±0.05 | 52.97±1.11 | 19.3 ± 3.25 | 10.3 ± 1.24 |
| w/ flat | 90.04±0.11 | 62.76±0.26 | 27.30±0.27 | 68.26±0.06 | 62.64±0.10 | 16.0 ± 2.37 | 8.03 ± 1.07 |
| w/ debias | 86.73±0.36 | 61.55±0.40 | 25.74±0.80 | 68.87±0.09 | 55.92±0.92 | 19.1 ± 3.45 | 8.44 ± 1.09 |
| w/ flat & debias | 89.34±0.40 | 61.90±0.43 | 27.41±0.28 | 67.64±0.15 | 62.57±0.22 | 14.6 ± 2.59 | 8.11 ± 1.09 |
| FastGCN (2) | 60.31±0.70 | 30.23±1.10 | 5.85±0.57 | 58.80±1.06 | 31.58±0.70 | 24.9 ± 5.09 | 8.34 ± 1.11 |
| LADIES (2) | 88.34±0.11 | 64.01±0.39 | 28.59±0.39 | 68.17±0.10 | 65.24±0.40 | 21.0 ± 4.00 | 11.2 ± 1.35 |
| w/ flat | 93.64±0.19 | **66.56±1.84** | 29.58±0.19 | 68.10±0.07 | 68.47±0.25 | 23.1 ± 4.04 | 13.1 ± 1.52 |
| w/ debias | 92.75±0.22 | 65.93±0.27 | **30.08±0.28** | **69.14±0.15** | 67.18±0.24 | 14.0 ± 1.94 | 8.54 ± 1.09 |
| w/ flat & debias | **93.59±0.09** | 66.22±0.10 | 29.88±0.34 | 67.75±0.11 | **68.49±0.06** | 21.0 ± 3.60 | 8.50 ± 1.15 |

The prediction accuracy (measured by corresponding metrics) on testing sets of different datasets is reported in Table 4. Our proposed methods, which combine the new sampling probabilities and the debiasing algorithm, are comparable to the full-batch training (no node sampling), showing consistent improvement over existing layer-wise sampling methods, FastGCN and LADIES. On most benchmarks, the prediction performance of our methods is better than the vanilla node-wise sampling method (GraphSAGE) and Graph-SAINT.

In addition, we would like to remark on the overlapping effect for the flat sampling probabilities and the debiasing algorithm. In Table 4, the accuracy of LADIES+flat+debias is better than LADIES+flat and LADIES+debias on most benchmarks, but this relative improvement is not remarkable on several benchmarks. This phenomenon is also observed in Figure 2 and Figure 3. The only exception is the ogbn-proteins dataset, where LADIES+flat+debias is inferior to LADIES+flat and LADIES+debias. However, on this dataset, LADIES and its variants even outperform "full-batch" GCN and VR-GCN. We tend to believe GCN's accuracy on ogbn-proteins is mainly impacted by factors other than the sampling variance.

We further remark on a seemingly strange phenomenon that some efficient GCNs have a higher prediction accuracy than full-batch GCN on several datasets. We speculate the reason is that a good approximation can recover the principal components in the original embedding matrix, restrain the noise via the sparse / low-rank structure, and serve as implicit regularization. There are similar observations (Sanyal et al., 2018; Chen et al., 2021) in Convolutional Neural Networks (CNN) and Transformers as well, that applying a low-rank regularizer, such as SVD, to the representation of the intermediate layers can improve the prediction accuracy of models.

Since LADIES improves upon FastGCN, alleviates the sparsity connection issue thereof, and performs consistently better than FastGCN on our benchmarks, the variants of FastGCN with our methods ("FastGCN+flat", "FastGCN+debias", and "FastGCN+flat+debias") are not the focus of this paper. We report their accuracy results in Appendix A for reference, where consistent improvements of "FastGCN+flat+debias" over FastGCN are observed on most benchmarks. We also notice that in some cases, the flat sampling probabilities and the debiasing algorithm bring insignificant improvements or even decreased accuracy. We conjugate that this is because FastGCN's structural deficiency somewhat distorts the convergence of GCN, as remarked by Zou et al. (2019). To be more specific, FastGCN does not apply a layer-dependent sampling strategy, which leads to the failure to capture the dynamics of mini-batch SGD and the overly sparse layer-wise connection. When the model does not converge well, the bias and variance of sampling is no longer the dominant factor in model's prediction accuracy.

### 6.3 Training Time

In the right-most columns of Table 4 [9], we report the training time for ogbn-arxiv and ogbn-product (complete runtime results for 2-layer and 3-layer GCNs are respectively provided in Tables 5 and 6 in Appendix A.3, along with the detailed timing settings). The difference between the runtime of our proposed methods and LADIES is not significant, since these layer-wise sampling strategies have the same propagation procedure. As for VR-GCN, a much heavier computational cost is required than the layer-wise methods as the expense of involving historical embedding, which helps it achieve the best accuracy on three tasks; in particular, its training time is even comparable to the full-batch method on ogbn-arxiv data.

Overall, we comment that based on the experimental results, our improvement in sampling probabilities and the proposed debiasing algorithm lead to better accuracy than the other two classical layer-wise sampling methods, FastGCN, and LADIES, while maintaining roughly the same computational cost.

## 7 Conclusion and Discussion

In this work, we revisit the existing layer-wise sampling strategies and propose two improvements. We first show that following the Principle of Maximum Entropy, a conservative choice of sampling probabilities outperforms the existing ones, whose proportionality assumption on embedding norms is in general unguaranteed. We further propose an efficient debiasing algorithm for layer-wise importance sampling through iterative updates of coefficients for columns sampled, and provide statistical analysis. The empirical experiments show our methods achieve high accuracy close to the SOTA node-wise sampling method, VR-GCN, while significantly saving runtime on GCN training like other history-oblivious layer-wise sampling methods.

We remark our debiased importance sampling strategy can be extended to a broader class of graph neural networks, such as node-wise sampling for GCN. Current node-wise sampling methods, e.g. GraphSAGE and VR-GCN, uniformly sample neighbors of each node *without* replacement. To further improve the approximation accuracy, node-wise sampling can also introduce importance sampling with our debiasing algorithm. Moreover, our debiasing algorithm can be applied to general machine learning involving sampling among a finite number of elements. In addition to the batch sampling for stochastic gradient descent (SGD) training discussed in Section 5.3, the proposed debiasing method can also contribute to sampling-based efficient attention, fast kernel ridge regression, etc.

### Acknowledgments

We appreciate all the valuable feedback from the anonymous reviewers and the TMLR editors. Y. Yang's research was supported in part by U.S. NSF grant DMS-2210717. R. Zhu's research was supported in part by U.S. NSF grant DMS-2210657.

---

[9]The least average training time is not boldfaced, since the training time on GPU is sensitive to the hardware and has a relatively large standard deviation the best result cannot significantly outperform the second-best one.

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

## A    Supplementary Experimental Setup and Results

### A.1    Experimental Setup Details

We describe the additional details of experiment setups for Section 6. All the models are implemented by PyTorch. We use one Tesla V100 SXM2 16GB GPU with 10 CPU threads to train all the models listed in Section 6. Our implementation of full-batch method, FastGCN, and LADIES are adapted from the codes by Zou et al. (2019); the implementation of vanilla node-wise sampling, VR-GCN, GraphSAINT is adapted from the codes by Cong et al. (2020). For the vanilla node-wise sampling method, there are several variants of GNN structures (Ying et al., 2018) while we fix the model structure as GCN in our experiments for fair comparison. We use ELU as the activation function in the convolutional layer for all the models: $ELU(x) = x$ for $x > 0$, $ELU(x) = \exp(x) - 1$ for $x \leq 0$.

For the details in model training, the learning rate is 0.001 and the dropout rate is 0.2, which means 20 percents of hidden units are randomly dropped during the training. Validation and testing are performed with full-batch inference (our experiments on those graph benchmarks are transductive: all the possible neighbors are accessible during training) on validation and testing nodes. Note that some existing PyTorch implementations of GCNs involve several ad-hoc tricks, such as row-normalizing sampled Laplacian matrix.

For the prediction accuracy evaluation experiments in Section 6, we stop training when the validation F1 score does not increase for 200 batches. For a fair comparison, we remove certain tricks in our experiments, such as normalization of each row in the sampled Laplacian matrix in layer-wise sampling. Such a trick may help in the practice, but it might not be compatible with some other methods and is out of the focus of our study. We use the metrics in Table 1 to evaluate the accuracy of each method. Concretely, Reddit is a multi-class classification task, and we use the Micro-F1 score with function "sklearn.metrics.f1_score". For OGB data, we use the built-in evaluator function in module `ogb` provided by Hu et al. (2020).

### A.2    Model Convergence

Figure 4 is a supplementary to Figure 3. We present the convergence curve of all methods on every task. The setting of each model is the same as in Figure 3.

### A.3    Sampling Time and Training Time

We compare the sampling time per batch for 1-layer GCN with layer-wise sampling methods (FastGCN, LADIES, and our proposed methods) and GraphSAGE by experiments on CPU. The time is presented in milliseconds. The batch size is 512, and the number of sampled nodes is 512 or 1024. The average sampling time (followed by standard deviation) over 200 batches is presented in Table 3. We note that the sampling time may involve some overhead costs. For example, the input Laplacian matrix is Scipy-spare-matrix on the CPU, while in sampling, it is converted to a PyTorch-sparse-matrix.

By Table 3, we conclude that the cost of debiasing algorithm is acceptable. Moreover, since the debiasing only depends on the number of nodes sampled, its time cost will be dwarfed by sampling on very large graphs. For example, sampling 512 nodes, the average batch sampling time for "LADIES", "LADIES + debiased", "LADIES + flat + debiased", are $8.3 \pm 0.1$, $11.7 \pm 0.2$, $11.7 \pm 0.5$ respectively on ogbn-arxiv while $83.4 \pm 1.3$ and $83.7 \pm 0.8$ and $83.5 \pm 0.6$ respectively on ogbn-products data. As we mentioned in Section 5.3, the node-wise sampling takes a significantly longer time because individually sampling from each row in the re-normalized Laplacian matrix (stored as a sparse matrix) leads to a large overhead cost.

We present the complete training time (per batch) of 2-layer and 3-layer GCNs in Tables 5 and 6 respectively. The time is presented in millisecond and averaged over 110 batches, where we discard the first 20 and the last 20 out of 150 total batches to disregard potential warm-up time for GPU. The other settings are kept the same as our experiments of accuracy evaluation in Table 4. We note that the timing on GPU is sensitive to the hardware and has a relatively large standard deviation.

As presented in Tables 5 and 6, our proposed methods have similar training time with LADIES due to the same propagation scheme of GCN with layer-wise sampling strategy. VR-GCN generally shows superiority

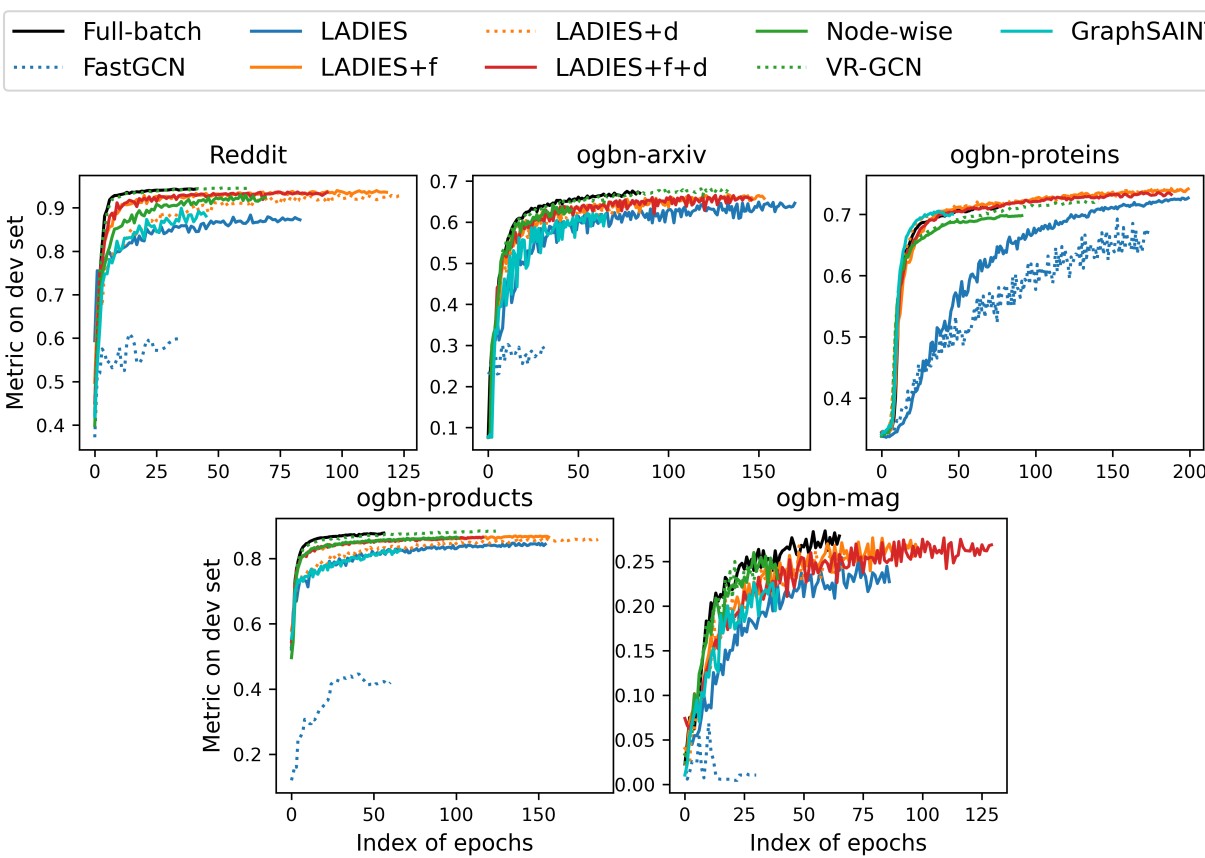

Figure 4: Metrics of each epoch on the validation set. The layer-wise sampling methods follow the "increasing" setting (denoted with a suffix (2) in Table 4); for the node-wise sampling methods, the number of neighbors is 2 per node.

in prediction accuracy (see Table 4). However, it also takes a significantly longer time in training since its propagation involves using and updating historical activation.

Table 5: Average training time (in milliseconds) per batch for a 2-layer GCN.

|                     | Reddit        | ogbn-arxiv      | ogbn-mag        | ogbn-proteins   | ogbn-products   |
| ------------------- | ------------- | --------------- | --------------- | --------------- | --------------- |
| Full-batch          | 372 ± 21.5    | 65.2 ± 3.97     | 72.3 ± 8.25     | 1702 ± 67.0     | 703 ± 77.8      |
| Node-wise (2)       | 8.13 ± 1.12   | 22.0 ± 3.64     | 17.3 ± 4.67     | 9.50 ± 1.29     | 8.34 ± 1.21     |
| Node-wise (10)      | 11.3 ± 1.05   | 19.7 ± 2.84     | 22.0 ± 5.60     | 10.3 ± 1.29     | 11.7 ± 1.31     |
| VR-GCN (2)          | 153 ± 17.3    | 86.8 ± 6.09     | 106 ± 12.4      | 239 ± 42.7      | 88.5 ± 2.51     |
| VR-GCN (10)         | 302 ± 23.9    | 104 ± 8.47      | 175 ± 15.1      | 360 ± 45.6      | 402 ± 65.3      |
| GraphSAINT          | 8.23 ± 1.14   | 23.1 ± 3.26     | 20.4 ± 5.98     | 8.34 ± 1.07     | 8.26 ± 1.11     |
| FastGCN             | 9.47 ± 1.21   | 24.1 ± 5.12     | 23.3 ± 6.27     | 8.50 ± 1.22     | 7.43 ± 1.04     |
| LADIES              | 7.95 ± 1.03   | 19.3 ± 3.25     | 16.7 ± 4.54     | 8.69 ± 1.11     | 10.3 ± 1.24     |
| w/ flat             | 7.86 ± 1.04   | 16.0 ± 2.37     | 26.1 ± 6.54     | 9.00 ± 1.21     | 8.03 ± 1.07     |
| w/ debiased         | 7.86 ± 1.11   | 19.1 ± 3.45     | 19.5 ± 5.67     | 8.21 ± 1.04     | 8.44 ± 1.09     |
| w/ flat & debiased  | 8.01 ± 1.09   | 14.6 ± 2.59     | 20.5 ± 5.62     | 9.06 ± 1.14     | 8.11 ± 1.09     |
| FastGCN (2)         | 9.47 ± 1.25   | 24.9 ± 5.09     | 20.8 ± 5.71     | 8.76 ± 1.06     | 8.34 ± 1.11     |
| LADIES (2)          | 14.2 ± 4.68   | 21.0 ± 4.00     | 22.7 ± 6.05     | 8.83 ± 1.12     | 11.2 ± 1.35     |
| w/ flat             | 8.70 ± 1.13   | 23.1 ± 4.04     | 21.9 ± 5.65     | 10.3 ± 1.24     | 13.1 ± 1.52     |
| w/ debiased         | 8.75 ± 1.15   | 14.0 ± 1.94     | 16.3 ± 4.64     | 10.7 ± 1.29     | 8.54 ± 1.09     |
| w/ flat & debiased  | 8.12 ± 1.13   | 21.0 ± 3.60     | 13.3 ± 4.15     | 12.5 ± 1.41     | 8.50 ± 1.15     |

Table 6: Average training time (in milliseconds) per batch for a 3-layer GCN.

|                     | Reddit         | ogbn-arxiv      | ogbn-mag        | ogbn-proteins    | ogbn-products     |
| ------------------- | -------------- | --------------- | --------------- | ---------------- | ----------------- |
| Full-batch          | 1042.8 ± 30.1  | 148 ± 5.8       | 352.1 ± 11.5    | 3312.3 ± 82.1    | 4490.7 ± 102.6    |
| Node-wise (2)       | 10.9 ± 1.3     | 27 ± 4.2        | 17.2 ± 4.3      | 11.5 ± 1.3       | 11.5 ± 1.1        |
| Node-wise (10)      | 77.8 ± 12      | 34.9 ± 3.3      | 52.4 ± 6.7      | 24.6 ± 1.1       | 50.4 ± 1.1        |
| VR-GCN (2)          | 379.7 ± 23.6   | 154.1 ± 12      | 218.7 ± 16.7    | 428.9 ± 53       | 473.6 ± 69.7      |
| VR-GCN (10)         | 858.1 ± 32     | 224.9 ± 17.8    | 488.1 ± 34.7    | 1618.7 ± 67.3    | 2075 ± 112.8      |
| GraphSAINT          | 9.5 ± 1.1      | 22.6 ± 3.2      | 21.2 ± 5.6      | 11.2 ± 1.3       | 10.5 ± 1.0        |
| FastGCN             | 16.9 ± 6.3     | 30.8 ± 4.3      | 19 ± 4.8        | 13 ± 1.3         | 8.9 ± 1.0         |
| LADIES              | 10.6 ± 1.3     | 27.6 ± 3.7      | 19.9 ± 4.8      | 11.1 ± 1.3       | 8.9 ± 1.0         |
| w/ flat             | 10.1 ± 1.1     | 26.4 ± 3.9      | 12.1 ± 2.4      | 10.1 ± 1.2       | 9.9 ± 1.0         |
| w/ debiased         | 10.1 ± 1.2     | 30 ± 4.1        | 22.5 ± 5.5      | 10.6 ± 1.2       | 10.6 ± 1.0        |
| w/ flat & debiased  | 9.7 ± 1.1      | 27.1 ± 3.6      | 15.7 ± 4.3      | 10.9 ± 1.2       | 9.3 ± 1.0         |
| FastGCN (2)         | 9.6 ± 1.2      | 27.4 ± 4.6      | 18.9 ± 4.7      | 9.8 ± 1.2        | 9.4 ± 1.1         |
| LADIES (2)          | 10 ± 1.2       | 29 ± 4.1        | 19.2 ± 5.1      | 10.5 ± 1.2       | 10.2 ± 1.1        |
| w/ flat             | 16.1 ± 4.9     | 27.2 ± 3.7      | 21.4 ± 5.8      | 10.4 ± 1.1       | 13.1 ± 1.4        |
| w/ debiased         | 10.3 ± 1.1     | 24.7 ± 3.1      | 23.3 ± 5.8      | 10.5 ± 1.2       | 12.4 ± 1.3        |
| w/ flat & debiased  | 10.7 ± 1.1     | 26.8 ± 4.4      | 24.8 ± 6.4      | 10.5 ± 1.1       | 9.9 ± 1.1         |

### A.4  Regression Experimental Setup

For the regression experiments in Section 4.1, we train a 3-layer GCN with LADIES sampler or full-batch sampler, with 256 hidden variables per layer. The batch size is 512. Early stopping training policy is applied. The regression lines in Figure 1 are fitted based on all the $(\|(\boldsymbol{HW})_{[i]}\|, \|\boldsymbol{P}^{[i]}\|)$ pairs collected from converged 3-layer GCN models in 5 repeated experiments. To make the pattern of the scatter plot clear, we only randomly sample 1000 pairs of data in each layer in each scatter plot. We also remark that these experiments are conducted to check the assumption of importance sampling, rather than pursuing SOTA performance. When we finish training the model, the norms of rows in $\boldsymbol{HW}$ are extracted through a full-batch inference with all training nodes.

We collect additional regression experiments in Appendix D.

Table 7: Testing metrics (detailed in Table 1) for FastGCN with flat sampling or/and debiasing on benchmarks. The metrics are in percentage (%).

| | Accuracy Metrics | | | | |
| | Reddit | ogbn-arxiv | ogbn-mag | ogbn-proteins | ogbn-products |
|---|---|---|---|---|---|
| LADIES | $73.86 \pm 0.17$ | $60.95 \pm 0.31$ | $24.79 \pm 0.48$ | $68.28 \pm 0.05$ | $52.97 \pm 1.11$ |
| FastGCN | $44.46 \pm 2.30$ | $25.44 \pm 0.82$ | $7.13 \pm 0.48$ | $52.44 \pm 1.88$ | $26.98 \pm 0.42$ |
| FastGCN w/ flat | $53.78 \pm 0.86$ | $22.76 \pm 0.69$ | $5.69 \pm 0.65$ | $61.68 \pm 0.50$ | $32.72 \pm 1.6$ |
| FastGCN w/ debias | $44.27 \pm 2.26$ | $26.57 \pm 2.52$ | $4.66 \pm 1.17$ | $53.64 \pm 1.33$ | $26.77 \pm 1.32$ |
| FastGCN w/ flat & debias | $53.88 \pm 0.94$ | $24.56 \pm 2.1$ | $7.54 \pm 0.31$ | $60.30 \pm 1.89$ | $29.79 \pm 2.19$ |
| LADIES (2) | $88.34 \pm 0.11$ | $64.01 \pm 0.39$ | $28.59 \pm 0.39$ | $68.17 \pm 0.10$ | $65.24 \pm 0.40$ |
| FastGCN (2) | $60.31 \pm 0.70$ | $30.23 \pm 1.10$ | $5.85 \pm 0.57$ | $58.80 \pm 1.06$ | $31.58 \pm 0.70$ |
| FastGCN (2) w/ flat | $59.60 \pm 1.07$ | $26.68 \pm 1.96$ | $6.68 \pm 0.03$ | $64.04 \pm 0.63$ | $35.27 \pm 1.14$ |
| FastGCN (2) w/ debias | $51.11 \pm 1.93$ | $25.65 \pm 1.02$ | $5.83 \pm 0.76$ | $56.30 \pm 2.49$ | $27.38 \pm 1.82$ |
| FastGCN (2) w/ flat & debias | $60.22 \pm 0.84$ | $24.41 \pm 1.02$ | $6.06 \pm 0.62$ | $63.13 \pm 0.57$ | $34.47 \pm 1.76$ |

### A.5 Supplementary Accuracy Results for FastGCN with flat sampling probabilities and debiasing

To complement the accuracy evaluation results in Table 4, we accordingly implement the variants of FastGCN ("FastGCN w/ flat", "FastGCN w/ debias", and "FastGCN w/ flat & debias") and perform the same experiments (all the five datasets in Section 6.2) on them. We still follow the same settings (including all the hyper-parameters in training) described at the beginning of Section 6: "FastGCN (2)" similarly refers to the "increasing" setting, where "double nodes will be sampled in the next layer". We summarize their accuracy results in Table 7, where we also provide the accuracy of original FastGCN and LADIES for comparison.

By introducing flat sampling probabilities and debasing algorithms, we observe "FastGCN w/ flat & debias" makes consistent improvements over FastGCN on most benchmarks. However, in some cases, the flat sampling probabilities and the debiasing algorithms bring insignificant improvements or even decreased accuracy. We provide a possible explanation for the phenomenon as follows. As illustrated in Table 7, FastGCN and its variants have significantly worse performance than even vanilla LADIES on all the benchmarks, which implies that the poor model convergence of FastGCN is mainly caused by a structural deficiency, the sparse layer-wise connection (Zou et al., 2019), on these benchmarks; regarding the impact on the model accuracy, the structural deficiency far outweighs the variance in sampling, and our improvement on sampling variance cannot always lead to higher model accuracy.

## B Time Complexity Analysis

Table 8: The time complexity for computation for $L$-layer GCN training by layer-wise sampling and node-wise sampling. The first column refers to the matrix operation type, nodes aggregation or linear transformation.

| | Layer-wise | Node-wise |
|---|---|---|
| Nodes Aggregation | $\mathcal{O}(scpL)$ | $\mathcal{O}(sb^L p)$ |
| Linear Transformation | $\mathcal{O}(sp^2 L)$ | $\mathcal{O}(sb^{L-1}p^2)$ |

We analyze the complexity of vanilla node-wise sampling and layer-wise sampling in this section. The analysis is adapted from the work by Zou et al. (2019), but we show a lighter bound for layer-wise sampling. For $l$ such that $0 \le l \le L - 1$, the propagation formulas for sampling based GCN can be formulated as:

$$\tilde{Z}^{(l+1)} = \bar{P}^{(l)} \tilde{H}^{(l)} W^{(l)},$$

where $\tilde{H}^{(l)} = \tilde{Z}^{(l)} \in \mathbb{R}^{s_l \times p}$, $\bar{P}^{(l)} \in \mathbb{R}^{s_{l+1} \times s_l}$, $W^{(l)} \in \mathbb{R}^{p \times p}$. In particular, for LADIES, $\bar{P}^{(l)}_{LADIES} = Q^{(l+1)} P S^{(l)}$.

For simplicity, we suppose that the hidden dimension in each layer is fixed as $p$, the same as the dimension of $H^{(0)}$. The batch size and the numbers of nodes sampled in each layer are set all equal to a fixed constant $s$.

We assume the number of sampled neighbors per node in node-wise sampling is $b$. We denote the maximal degree of all the nodes in the graph as $c$. The computational cost of the propagation comes from two parts: the linear transformation, a dense matrix product, $\tilde{\boldsymbol{H}}^{(l)}\boldsymbol{W}^{(l)}$ and the node aggregation, a sparse matrix product, $\bar{\boldsymbol{P}}^{(l)}(\tilde{\boldsymbol{H}}^{(l)}\boldsymbol{W}^{(l)})$. The time complexity is summarized in Table 8. We additionally comment although the time cost of two parts both linearly depend on the number of nodes involved (number of non-zero elements in $\boldsymbol{Q}^{(l+1)}$), the node aggregation part usually dominates since the sparse matrix product involved is less efficient than the dense matrix product involved in a modern computer.

The linear transformation, $\tilde{\boldsymbol{H}}^{(l)}\boldsymbol{W}^{(l)}$ is dense matrix production. The cost depends on the shape of two matrices, and is given as $\mathcal{O}(s_l p^2)$. LADIES fixes $s_l$ as $s$ for each layer, so $\mathcal{O}(s_l p^2) = \mathcal{O}(sp^2)$. For node-wise sampling $s_l = sb^{L-l}$, since the number of node grows exponentially. Thus, by summation over all the layers, we have the results in the second row in Table 8.

The node aggregation, $\bar{\boldsymbol{P}}^{(l)}(\tilde{\boldsymbol{H}}^{(l)}\boldsymbol{W}^{(l)})$ is a sparse matrix production, since $\bar{\boldsymbol{P}}^{(l)}$ is sparse. For simplicity, we denote $\tilde{\boldsymbol{H}}^{(l)}\boldsymbol{W}^{(l)}$ as $\boldsymbol{C}^{(l)} \in \mathbb{R}^{s_l \times p}$. Thus, the time complexity of this sparse matrix production becomes $\mathcal{O}(\text{nnz}^{(P_l)}p)$, where $\text{nnz}^{(P_l)}$ is the number of non-zero entries in $\bar{\boldsymbol{P}}^{(l)}$. For layer-wise sampling, since we sample $s$ nodes for each layer and each node has at most $c$ neighbors, so $\text{nnz}^{(P_l)} \le sc$. For node-wise sampling, since each node has $b$ neighbors and the neighbors are not shared by all the nodes in each layer, $\text{nnz}^{(P_l)} = bs_l = sb^{L+1-l}$. By summation over all the layers, we attain the results in the first row of Table 8.

## C Theoretical Analysis of Sampling Probabilities

### C.1 Results in approximate matrix multiplication

In this section, we revisit approximate matrix multiplication to derive the previous layer-wise sampling methods. Specifically, the sampling matrix $\boldsymbol{S}$ used in FastGCN and LADIES can be decomposed as $\boldsymbol{S} = \boldsymbol{\Pi}\boldsymbol{\Pi}^T$, where $\boldsymbol{\Pi} \in \mathbb{R}^{n \times d}$ is a sub-sampling sketching matrix defined as follows:

**Definition C.1** (Sub-sampling sketching matrix). *Consider a discrete distribution which draws $i$ with probability $p_i > 0, \forall i \in [n]$. For a random matrix $\boldsymbol{\Pi} \in \mathbb{R}^{n \times d}$, if $\boldsymbol{\Pi}$ has i.i.d. columns and each column $\boldsymbol{\Pi}^{(j)}$ can randomly be $\frac{1}{\sqrt{dp_i}}\boldsymbol{e}_i$ with probability $p_i$, where $\boldsymbol{e}_i$ is the $i$-th column of the $n$-by-$n$ identity matrix $\boldsymbol{I}_n$, then $\boldsymbol{\Pi}$ is called a sub-sampling sketching matrix with sub-sampling probabilities $\{p_i\}_{i=1}^n$.*

With this definition, we introduce a result in AMM to construct the sub-sampling sketching matrix, which coincides with the conclusion in FastGCN and LADIES.

**Theorem C.1** (Theorem 1 (Drineas et al., 2006)). *Suppose $\boldsymbol{B} \in \mathbb{R}^{n_B \times n}$, $\boldsymbol{C} \in \mathbb{R}^{n \times n_C}$, the number of sub-sampled columns $d \in \mathbb{Z}^+$ such that $1 \le d \le n$, and the sub-sampling probabilities $\{p_i\}_{i=1}^n$ are such that $\sum_{i=1}^n p_i = 1$ and such that for a quality coefficient $\beta \in (0, 1]$*

$$p_i \ge \beta \frac{\|\boldsymbol{B}^{[i]}\|\|\boldsymbol{C}_{[i]}\|}{\sum_{i'=1}^n \|\boldsymbol{B}^{[i']}\|\|\boldsymbol{C}_{[i']}\|}, \forall i \in [n]. \tag{13}$$

*Construct a sub-sampling sketching matrix $\boldsymbol{\Pi} \in \mathbb{R}^{n \times d}$ with sub-sampling probabilities $\{p_i\}_{i=1}^n$ as in Definition C.1, and let $\boldsymbol{B}\boldsymbol{\Pi}\boldsymbol{\Pi}^T\boldsymbol{C}$ be an approximation to $\boldsymbol{B}\boldsymbol{C}$. Let $\delta \in (0, 1)$ and $\eta = 1 + \sqrt{(8/\beta)\log(1/\delta)}$. Then with probability at least $1 - \delta$,*

$$\|\boldsymbol{B}\boldsymbol{C} - \boldsymbol{B}\boldsymbol{\Pi}\boldsymbol{\Pi}^T\boldsymbol{C}\|_F^2 \le \frac{\eta^2}{\beta d}\|\boldsymbol{B}\|_F^2\|\boldsymbol{C}\|_F^2. \tag{14}$$

**Remark.** The theorem is closely related to Lemma 1 in Appendix B of LADIES, which studies the variance $\mathbb{E}\|\boldsymbol{B}\boldsymbol{C} - \boldsymbol{B}\boldsymbol{S}\boldsymbol{C}\|_F^2$. For the choice of sub-sampling probabilities, Equation (13) reproduces the conclusion in FastGCN and LADIES, when we respectively take $\boldsymbol{B}$ as $\boldsymbol{P}$ and $\boldsymbol{Q}\boldsymbol{P}$.

### C.2 Comparison of Sampling Variance Between Our Sampling Probabilities and LADIES

Whether our choice of probabilities can outperform the LADIES depends on the distribution of the norms of rows in $\boldsymbol{H}\boldsymbol{W}$. When $\|\boldsymbol{H}\boldsymbol{W}_{(i)}\|$ is not proportional to the corresponding $\ell_2$ norm of column $(\boldsymbol{Q}\boldsymbol{P})^{(i)}$, our

proposed probabilities can benefit the approximate matrix multiplication task more than the ones assuming a relation of proportionality. We find the common long-tail distribution of numbers suffices to exert the strengths of the new probabilities, which can be concluded as the following assumption:

**Assumption 1.** *To simplify the notation, we denote $\boldsymbol{B} := \boldsymbol{QP}$ and $\boldsymbol{C} := \boldsymbol{HW}$, where $\boldsymbol{P}$ is an n-by-n matrix as defined above. Let $m$ be the number of non-zero columns in $\boldsymbol{B}$, and define $C_1 := \frac{\|\boldsymbol{B}\|_F^2/m}{\left(\sum_{i=1}^n \|\boldsymbol{B}^{[i]}\|/m\right)^2} \geq 1$. There also exists a constant $C_2 \geq 1$ such that $\frac{1}{C_2}\|\boldsymbol{C}\|_F^2/n \leq \|\boldsymbol{C}_{[i]}\|^2 \leq C_2\|\boldsymbol{C}\|_F^2/n$. Assume $C_1/C_2^2 \geq 1$.*

With the assumption above, we show the variance of the approximation with our proposed probabilities is smaller than the variance of LADIES by the following lemma.

**Lemma C.1.** *We denote the sampling matrix with our probabilities in Equation (7) as $\boldsymbol{S}_1$, and denote the sampling matrix with probabilities of LADIES in Equation (2) as $\boldsymbol{S}_0$. If Assumption 1 holds, then we have*

$$\mathbb{E}\,\|\boldsymbol{BS}_1\boldsymbol{C} - \boldsymbol{BC}\|_F^2 \leq \mathbb{E}\,\|\boldsymbol{BS}_0\boldsymbol{C} - \boldsymbol{BC}\|_F^2.$$

**Remark.** Assumption 1 is related to the uniformity in the distributions of $\|\boldsymbol{B}^{[i]}\|$'s and $\|\boldsymbol{C}_{[i]}\|$'s. We tentatively discuss the implication of the assumption in Appendix C.2. We remark the assumption indicates it is unrealistic that the new probabilities can outperform the ones in LADIES, as distributions of datasets can vary. Nevertheless, as shown in Section 6 it can be an effective attempt to improve the prediction accuracy of LADIES by simply adopting the conservative sampling scheme.

To prove Lemma C.1, we first adapt a technical lemma (Zou et al., 2019, Lemma 1), which relates the sampling matrix to the variance (expectation of squared Frobenius norm) of the approximate matrix multiplication.

**Lemma C.2** (Adapted from Lemma 1 (Zou et al., 2019)). *Given two matrices $\boldsymbol{B} \in \mathbb{R}^{n_B \times n}$ and $\boldsymbol{C} \in \mathbb{R}^{n \times n_C}$, for any $i \in [n]$ define the positive probabilities $p_i$'s such that $\sum_{i=1}^n p_i = 1$. We further require the probability $p_i = 0$ if and only if the corresponding column $\boldsymbol{B}^{[i]}$ or row $\boldsymbol{C}_{[i]}$ is all-zero. The sub-sampling sketching matrix $\boldsymbol{\Pi} \in \mathbb{R}^{n \times d}$ is generated accordingly. Let $\boldsymbol{S} := \boldsymbol{\Pi}\boldsymbol{\Pi}^T$, it holds that*

$$\mathbb{E}_{\boldsymbol{S}}\left[\|\boldsymbol{BSC} - \boldsymbol{BC}\|_F^2\right] = \frac{1}{d}\left(\sum_{i:p_i>0}\frac{1}{p_i}\left\|\boldsymbol{B}^{[i]}\right\|^2 \cdot \left\|\boldsymbol{C}_{[i]}\right\|^2 - \|\boldsymbol{BC}\|_F^2\right)$$

*where $d$ is the number of samples.*

With the lemma above, the proof of Lemma C.1 is provided as follows.

*Proof.* Recall the notation in the main paper is simplified as $\boldsymbol{B} := \boldsymbol{QP}, \boldsymbol{C} := \boldsymbol{HW}$. As the union of neighbors of nodes in $\boldsymbol{Q}$ cannot cover all the nodes, some columns in $\boldsymbol{B}$ are all-zero, and we accordingly define a $\boldsymbol{Q}$-measurable matrix $\boldsymbol{L}$ as in Lemma C.2. We have

$$\mathbb{E}\left[\|\boldsymbol{BS_1}\boldsymbol{C} - \boldsymbol{BC}\|_F^2\right] = \mathbb{E}_{\boldsymbol{Q}}\left[\mathbb{E}_{\boldsymbol{S}_1}\left(\|\boldsymbol{BS_1}\boldsymbol{C} - \boldsymbol{BC}\|_F^2|\boldsymbol{Q}\right)\right]$$

$$= \frac{1}{d}\mathbb{E}_{\boldsymbol{Q}}\left[\sum_{i:p_i>0}\frac{1}{p_i}\left\|\boldsymbol{B}^{[i]}\right\|^2 \cdot \left\|\boldsymbol{C}_{[i]}\right\|^2 - \|\boldsymbol{BC}\|_F^2\right].$$

where the second equation holds as we apply Lemma C.2 to the inner expectation in the right-hand side of the first line. Plugging $p_i \propto \|\boldsymbol{B}^{[i]}\|$ (Equation (7) in the main paper) into the preceding probabilities $p_i$'s, we reach

$$\mathbb{E}\left[\|\boldsymbol{BS_1}\boldsymbol{C} - \boldsymbol{BC}\|_F^2\right] = \frac{\mathbb{E}_{\boldsymbol{Q}}\left[\left(\sum_{i:p_i>0}\|\boldsymbol{B}^{[i]}\|\right)\left(\sum_{i:p_i>0}\|\boldsymbol{B}^{[i]}\|\,\|\boldsymbol{C}_{[i]}\|^2\right)\right]}{d} - \frac{\mathbb{E}_{\boldsymbol{Q}}\left[\|\boldsymbol{BC}\|_F^2\right]}{d}$$

$$= \frac{1}{d}\mathbb{E}_{\boldsymbol{Q}}\left[\left(\sum_{i=1}^n\left\|\boldsymbol{B}^{[i]}\right\|\right)\left(\sum_{i=1}^n\left\|\boldsymbol{B}^{[i]}\right\|\,\|\boldsymbol{C}_{[i]}\|^2\right)\right] - \frac{1}{d}\mathbb{E}_{\boldsymbol{Q}}\left[\|\boldsymbol{BC}\|_F^2\right].$$

As computed by Zou et al. (2019), the variance of LADIES is similarly given as

$$\mathbb{E}\left[\|\boldsymbol{B}\boldsymbol{S_0}\boldsymbol{C} - \boldsymbol{B}\boldsymbol{C}\|_F^2\right] = \frac{1}{d}\,\mathbb{E}_{\boldsymbol{Q}}\left[\left(\sum_{i:p_i>0}\left\|\boldsymbol{B}^{[i]}\right\|^2\right)\left(\sum_{i:p_i>0}\left\|\boldsymbol{C}_{[i]}\right\|^2\right)\right] - \frac{1}{d}\,\mathbb{E}_{\boldsymbol{Q}}\left[\|\boldsymbol{B}\boldsymbol{C}\|_F^2\right].$$

Consequently, to prove the lemma it suffices to show that

$$\left(\sum_{i:p_i>0}\left\|\boldsymbol{B}^{[i]}\right\|\right)\left(\sum_{i:p_i>0}\left\|\boldsymbol{B}^{[i]}\right\|\left\|\boldsymbol{C}_{[i]}\right\|^2\right) \le \left(\sum_{i:p_i>0}\left\|\boldsymbol{B}^{[i]}\right\|^2\right)\left(\sum_{i:p_i>0}\left\|\boldsymbol{C}_{[i]}\right\|^2\right), \qquad (15)$$

and the inequality above follows with Assumption 1. Specifically, plugging the inequality $\|\boldsymbol{C}_{[i]}\|^2 \le C_2\|\boldsymbol{C}\|_F^2/n, \forall i \in [n]$ in the left-hand-side above, we have

$$\left(\sum_{i:p_i>0}\left\|\boldsymbol{B}^{[i]}\right\|\right)\left(\sum_{i:p_i>0}\left\|\boldsymbol{B}^{[i]}\right\|\left\|\boldsymbol{C}_{[i]}\right\|^2\right) \le \left(\sum_{i=1}^{n}\|\boldsymbol{B}^{[i]}\|\right)^2\frac{C_2}{n}\|\boldsymbol{C}\|_F^2 = \frac{m}{C_1}\|\boldsymbol{B}\|_F^2\frac{C_2}{nm}m\|\boldsymbol{C}\|_F^2,$$

in which the last equation comes from the definition $C_1 := \frac{\|\boldsymbol{B}\|_F^2/m}{\left(\sum_{i=1}^{n}\|\boldsymbol{B}^{[i]}\|/m\right)^2}$. To close the proof, we utilize the inequality $\frac{1}{C_2}\|\boldsymbol{C}\|_F^2/n \le \|\boldsymbol{C}_{[i]}\|^2$ and bound $m\|\boldsymbol{C}\|_F^2$ by $nC_2\sum_{i:p_i>0}\left\|\boldsymbol{C}_{[i]}\right\|^2$. Finally we attain Equation (15) with the core assumption $\frac{C_1}{C_2^2} \ge 1$. $\diamondsuit$

**Remark.** In Assumption 1 we indeed implicitly assume $\|\boldsymbol{B}^{[i]}\|$'s follow a long-tail distribution that most norms are around the average while a few columns have large norms. The high non-uniformity makes the average of squared norms much larger than the square of averaged norms. For $\|\boldsymbol{C}_{[i]}\|$'s, considering the normalization techniques (such as batch or layer normalization) to stabilize the scale of the parameters, they tend to not vary widely, which implies a small $C_2$. The numerical experiments on the comparison of approximation error (see Figure 2) and the histograms of the norms in trained models shown in Figure 5 further validate the assumption. Based on the empirical analysis above, we claim the assumption is mild and tends to hold at least for some datasets.

### C.3 Distributions of the Matrix Rows / Columns Norm

Figure 5 demonstrates the distribution of $\|\boldsymbol{P}^{[i]}\|$ and $\|(\boldsymbol{H}\boldsymbol{W})_{[i]}\|$ (layer 1, 2, 3) for Reddit, ogbn-arxiv, ogbn-proteins, and ogbn-mag datasets. The $\|(\boldsymbol{H}\boldsymbol{W})_{[i]}\|$'s are obtained from the experiment in Section A.4. The outliers larger than the 99.9% quantile or small than the 0.1% quantile are removed.

As shown in the histograms, our analysis regarding Assumption 1 tends to hold generally on these datasets. For the norms of columns in $\boldsymbol{P}$ (as a replacement for $\boldsymbol{Q}\boldsymbol{P}$ for clarity), we observe there are some columns with large norms far beyond the average. Those columns contribute a lot to the quadratic mean, which results in a huge $C_1$ in Assumption 1. In contrast, the norms of rows in $\boldsymbol{H}\boldsymbol{W}$ concentrate around their average, inducing a small $C_2$. Those facts together with Assumption 1 and Lemma C.1 explain why our proposed sampling probabilities are more proper for some real datasets.

### C.4 Proof of Theorem 1

*Proof.* We first show $\mathbb{E}\,\beta_i^{(k)} = 1, \forall i \in [n], k \in [s]$. As $\beta_i^{(k)}$ is constructed by Algorithm 1 to attain the unbiased estimator, take $\boldsymbol{X}_i = 1, \boldsymbol{X}_j = 0, \forall j \ne i$, and we have $\mathbb{E}\,\beta_i^{(k)} = \mathbb{E}\sum_{j=1}^{n}\beta_j^{(k)}\boldsymbol{X}_j = \sum_{j=1}^{n}\boldsymbol{X}_j = 1, \forall i \in [n], k \in [s]$.

With $\mathbb{E}\,\beta_i^{(k+1)}$ at hand, we still need to compute $\mathbb{E}\,(\beta_i^{(k+1)})^2$ (and $\mathbb{E}\,\beta_i^{(k+1)}\beta_j^{(k+1)}$) to obtain the (co)variance. To start the analysis, we recursively write $\beta_i^{(k+1)}$ as

$$\beta_i^{(k+1)} = \mathbf{1}_{\{i\in S_k\}}[\beta_i^{(k)}(1-\alpha_{k+1}) + \alpha_{k+1}] + \mathbf{1}_{\{i\notin S_k\}}\mathbf{1}_{\{i\in S_{k+1}\}}\frac{1-\sum_{j\in S_k}p_j}{p_i}\alpha_{k+1} := \pi_i^{k+1}(\beta_i^{(k)}) + \gamma_i^{(k+1)}.$$

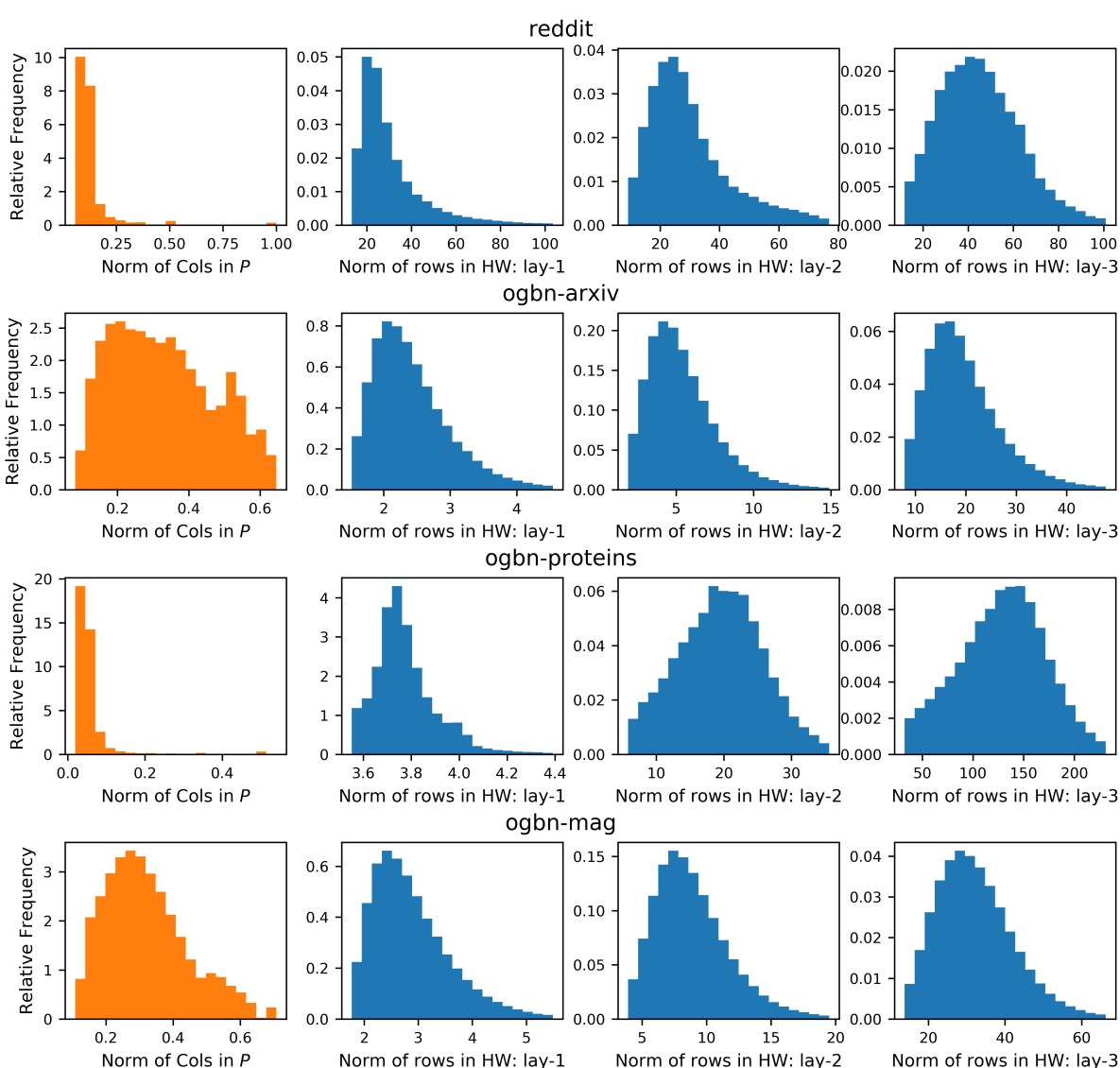

Figure 5: Distributions of $\|\boldsymbol{P}^{[i]}\|$'s and $\|(\boldsymbol{HW})_{[i]}\|$'s for Reddit, ogbn-arxiv, ogbn-protein and ogbn-mag.

For $\mathbb{E}\,(\beta_i^{(k+1)})^2$, we notice the cross term $2\pi_i^{k+1}(\beta_i^{(k)})\gamma_i^{(k+1)}$ is always zero as $\mathbf{1}_{\{i\in S_k\}}\mathbf{1}_{\{i\notin S_k\}} := 0$; as for the first terms, utilizing the fact $\mathbf{1}_{\{i\in S_k\}}\beta_i^{(k)} = \beta_i^{(k)}$ we have

$$\mathbb{E}\left(\pi_i^{k+1}(\beta_i^{(k)})\right)^2 = \mathbb{E}\,(\beta_i^{(k)})^2(1-\alpha_{k+1})^2 + 2\alpha_{k+1}(1-\alpha_{k+1}) + q_i^k\alpha_{k+1}^2,$$

to obtain the last term,

$$\begin{aligned}
\mathbb{E}\,(\gamma_i^{(k+1)})^2 &= \mathbb{E}\,\mathbb{E}\left(\mathbf{1}_{\{i\notin S_k\}}\mathbf{1}_{\{i\in S_{k+1}\}}\frac{1-\sum_{j\in S_k}p_j}{p_i}\alpha_{k+1}|i\notin S_k\right) \\
&= \mathbb{E}\left[\mathbf{1}_{\{i\notin S_k\}}\mathbb{E}\left(\mathbf{1}_{\{i\in S_{k+1}\}}\frac{1-\sum_{j\in S_k}p_j}{p_i}\alpha_{k+1}|i\notin S_k\right)\right] \\
&= q_i^k\sum_{S_k\not\ni i}\frac{p_{S_k}}{q_i^k}\frac{1-\sum_{j\in S_k}p_j}{p_i}\alpha_{k+1}^2 = \frac{r_i^k}{p_i}\alpha_{k+1}^2.
\end{aligned}$$

For $\mathbb{E}\,\beta_i^{(k+1)}\beta_j^{(k+1)}$, we can similarly drop the last term $\mathbb{E}\,\gamma_i^{(k+1)}\gamma_j^{(k+1)}$ as $\mathbf{1}_{\{i\in S_k\}}\mathbf{1}_{\{i\notin S_k\}} := 0$; as for the first term $\mathbb{E}\,\pi_i^{k+1}(\beta_i^{(k)})\pi_j^{k+1}(\beta_j^{(k)})$, we have

$$\mathbb{E}\,\pi_i^{k+1}(\beta_i^{(k)})\pi_j^{k+1}(\beta_j^{(k)}) = \mathbb{E}\,\beta_i^{(k)}\beta_j^{(k)}(1-\alpha_{k+1})^2 + \mathbb{E}\left(\mathbf{1}_{\{i\in S_k\}}\beta_j^{(k)} + \mathbf{1}_{\{j\in S_k\}}\beta_i^{(k)}\right)\alpha_{k+1}(1-\alpha_{k+1}) + p_{i,j}^k\alpha_{k+1}^2,$$

where $p_{i,j}^k$ is the probability that **both** index $i,j$ are in the first k samples; as for the next term $\mathbb{E}\,\pi_i^{k+1}(\beta_i^{(k)})\gamma_j^{(k+1)}$, we first compute

$$\begin{aligned}
\mathbb{E}\,\beta_i^{(k)}\gamma_j^{(k+1)} &= \mathbb{E}\,\mathbb{E}\left(\beta_i^{(k)}\mathbf{1}_{\{j\notin S_k\}}\mathbf{1}_{\{j\in S_{k+1}\}}\frac{1-\sum_{j'\in S_k}p_{j'}}{p_j}\alpha_{k+1}|\mathcal{F}_k\right) \\
&= \mathbb{E}\left[\beta_i^{(k)}\mathbf{1}_{\{j\notin S_k\}}\frac{1-\sum_{j'\in S_k}p_{j'}}{p_j}\alpha_{k+1}\,\mathbb{E}\left(\mathbf{1}_{\{j\in S_{k+1}\}}|\mathcal{F}_k\right)\right] \\
&= \mathbb{E}\left[\beta_i^{(k)}\mathbf{1}_{\{j\notin S_k\}}\frac{1-\sum_{j'\in S_k}p_{j'}}{p_j}\alpha_{k+1}\frac{p_j}{1-\sum_{j'\in S_k}p_{j'}}\right] = \mathbb{E}\left(\mathbf{1}_{\{j\notin S_k\}}\beta_i^{(k)}\right)\alpha_{k+1},
\end{aligned}$$

and similarly we have

$$\begin{aligned}
\mathbb{E}\,\mathbf{1}_{\{i\in S_k\}}\gamma_j^{(k+1)} &= \mathbb{E}\,\mathbb{E}\left(\mathbf{1}_{\{i\in S_k\}}\mathbf{1}_{\{j\notin S_k\}}\mathbf{1}_{\{j\in S_{k+1}\}}\frac{1-\sum_{j'\in S_k}p_{j'}}{p_j}\alpha_{k+1}|\mathcal{F}_k\right) \\
&= \mathbb{E}\left[\mathbf{1}_{\{i\in S_k\}}\mathbf{1}_{\{j\notin S_k\}}\frac{1-\sum_{j'\in S_k}p_{j'}}{p_j}\alpha_{k+1}\,\mathbb{E}\left(\mathbf{1}_{\{j\in S_{k+1}\}}|\mathcal{F}_k\right)\right] \\
&= \mathbb{E}\left[\mathbf{1}_{\{i\in S_k\}}\mathbf{1}_{\{j\notin S_k\}}\frac{1-\sum_{j'\in S_k}p_{j'}}{p_j}\alpha_{k+1}\frac{p_j}{1-\sum_{j'\in S_k}p_{j'}}\right] = \mathbb{E}\left(\mathbf{1}_{\{j\notin S_k\}}\mathbf{1}_{\{i\in S_k\}}\right)\alpha_{k+1};
\end{aligned}$$

accordingly we can obtain

$$\mathbb{E}\,\pi_i^{k+1}(\beta_i^{(k)})\gamma_j^{(k+1)} = \mathbb{E}\left(\mathbf{1}_{\{j\notin S_k\}}\beta_i^{(k)}\right)\alpha_{k+1}(1-\alpha_{k+1}) + \mathbb{E}\left(\mathbf{1}_{\{j\notin S_k\}}\mathbf{1}_{\{i\in S_k\}}\right)\alpha_{k+1}^2,$$

and applying the same derivation as above we have

$$\mathbb{E}\,\pi_j^{k+1}(\beta_j^{(k)})\gamma_i^{(k+1)} = \mathbb{E}\left(\mathbf{1}_{\{i\notin S_k\}}\beta_j^{(k)}\right)\alpha_{k+1}(1-\alpha_{k+1}) + \mathbb{E}\left(\mathbf{1}_{\{i\notin S_k\}}\mathbf{1}_{\{j\in S_k\}}\right)\alpha_{k+1}^2.$$

Combining all the pieces together, we obtain

$$\begin{aligned}
\mathbb{E}\,\beta_i^{(k+1)}\beta_j^{(k+1)} &= \mathbb{E}\,\beta_i^{(k)}\beta_j^{(k)}(1-\alpha_{k+1})^2 + 2\alpha_{k+1}(1-\alpha_{k+1}) + \left(p_{i,j}^k + \mathbb{E}\left(\mathbf{1}_{\{j\notin S_k\}}\mathbf{1}_{\{i\in S_k\}} + \mathbf{1}_{\{i\notin S_k\}}\mathbf{1}_{\{j\in S_k\}}\right)\right)\alpha_{k+1}^2 \\
&= \mathbb{E}\,\beta_i^{(k)}\beta_j^{(k)}(1-\alpha_{k+1})^2 + 2\alpha_{k+1}(1-\alpha_{k+1}) + \left(p_{i,j}^k + \mathbb{E}\left(\mathbf{1}_{\{j\notin S_k\}}\mathbf{1}_{\{i\in S_k\}} + \mathbf{1}_{\{i\notin S_k\}}\mathbf{1}_{\{j\in S_k\}}\right)\right)\alpha_{k+1}^2.
\end{aligned}$$

With the derivation above, we have

$$\mathbb{E}\,(\beta_i^{(k+1)})^2 = \mathbb{E}\,(\beta_i^{(k)})^2(1-\alpha_{k+1})^2 + 2\alpha_{k+1}(1-\alpha_{k+1}) + (\frac{r_i^{(k)}}{p_i} + q_i^k)\alpha_{k+1}^2,$$

$$\mathbb{E}\,\beta_i^{(k+1)}\beta_j^{(k+1)} = \mathbb{E}\,\beta_i^{(k)}\beta_j^{(k)}(1-\alpha_{k+1})^2 + 2\alpha_{k+1}(1-\alpha_{k+1}) + q_{i,j}^k\alpha_{k+1}^2$$

where $q_i^k(= 1 - \bar{q}_i^{(k)})$ is the probability that index $i$ is in the first $k$ samples, and similarly $q_{i,j}^k(= 1 - \bar{q}_{i,j}^{(k)} = q_i^k + q_j^k - p_{i,j}^k)$ is the probability that either index $i$ or index $j$ is in the first $k$ samples. Plugging the expression above into the following identities,

$$\mathrm{Var}(\beta_i^{(k+1)}) = \mathbb{E}\,(\beta_i^{(k+1)})^2 - \mathbb{E}^2\,(\beta_i^{(k+1)})$$
$$\mathrm{Cov}(\beta_i^{(k+1)}, \beta_j^{(k+1)}) = \mathbb{E}\,\beta_i^{(k+1)}\beta_j^{(k+1)} - \mathbb{E}\,\beta_i^{(k+1)}\,\mathbb{E}\,\beta_j^{(k+1)},$$

we can then have the expression for the covariance stated in the main paper.

For the scale of the covariance, we prove the upper bound through induction. We can verify the upper bounds hold for $k = 1$, and for the (co)variance with $\alpha_k = \frac{1}{k}$, $\mathrm{Var}(\beta_i^{(k+1)})$ and $\mathrm{Cov}(\beta_i^{(k+1)}, \beta_j^{(k+1)})$ now respectively becomes

$$\mathrm{Var}(\beta_i^{(k+1)}) = \frac{k^2}{(k+1)^2}\mathrm{Var}(\beta_i^{(k)}) + \left(\frac{r_i^{(k)}}{p_i} - \bar{q}_i^{(k)}\right)\frac{1}{(k+1)^2},$$

$$\mathrm{Cov}(\beta_i^{(k+1)}, \beta_j^{(k+1)}) = \frac{k^2}{(k+1)^2}\mathrm{Cov}(\beta_i^{(k)}, \beta_j^{(k)}) - \bar{q}_{i,j}^{(k)}\frac{1}{(k+1)^2}.$$

Utilizing the induction conditions that for all $i, j$, we have

$$\mathrm{Var}(\beta_i^{(k)}) \leq \frac{1}{k}(\frac{1}{p_i} - 1),$$

$$|\mathrm{Cov}(\beta_i^{(k)}, \beta_j^{(k)})| \leq \frac{1}{k}.$$

Along with the facts that $\frac{r_i^{(k)}}{p_i} - \bar{q}_i^{(k)} \leq \frac{\bar{q}_i^{(k)}}{p_i} - \bar{q}_i^{(k)} \leq \frac{1}{p_i} - 1$ and $\bar{q}_{i,j}^{(k)} \leq 1$, we can finally achieve the inequality that

$$\mathrm{Var}(\beta_i^{(k+1)}) \leq \frac{1}{k+1}(\frac{1}{p_i} - 1),$$

$$|\mathrm{Cov}(\beta_i^{(k+1)}, \beta_j^{(k+1)})| \leq \frac{1}{k+1}.$$

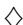

**Remark.** The choice of $\alpha_k = \frac{1}{k}$ here is mainly for easing the proof, while may not be the optimal choice in practice; indeed in SRS the $\alpha_k$'s are different than the ones used here.

# D Supplementary Regression Results

## D.1 Full Batch Training

In this subsection, we present the regression analysis for GCN with full-batch SGD training (without sampling). Figure 6 shows a similar pattern, as supplementary to Figure 1. Here, the scatter plots make use of all points. The assumption: $\|(HW)_{[i]}\| \propto \|P^{[i]}\|$ still does not hold. Note that we do not have the regression result on the ogbn-product dataset, since the training of a 3-layer GCN fails due to memory limitation.

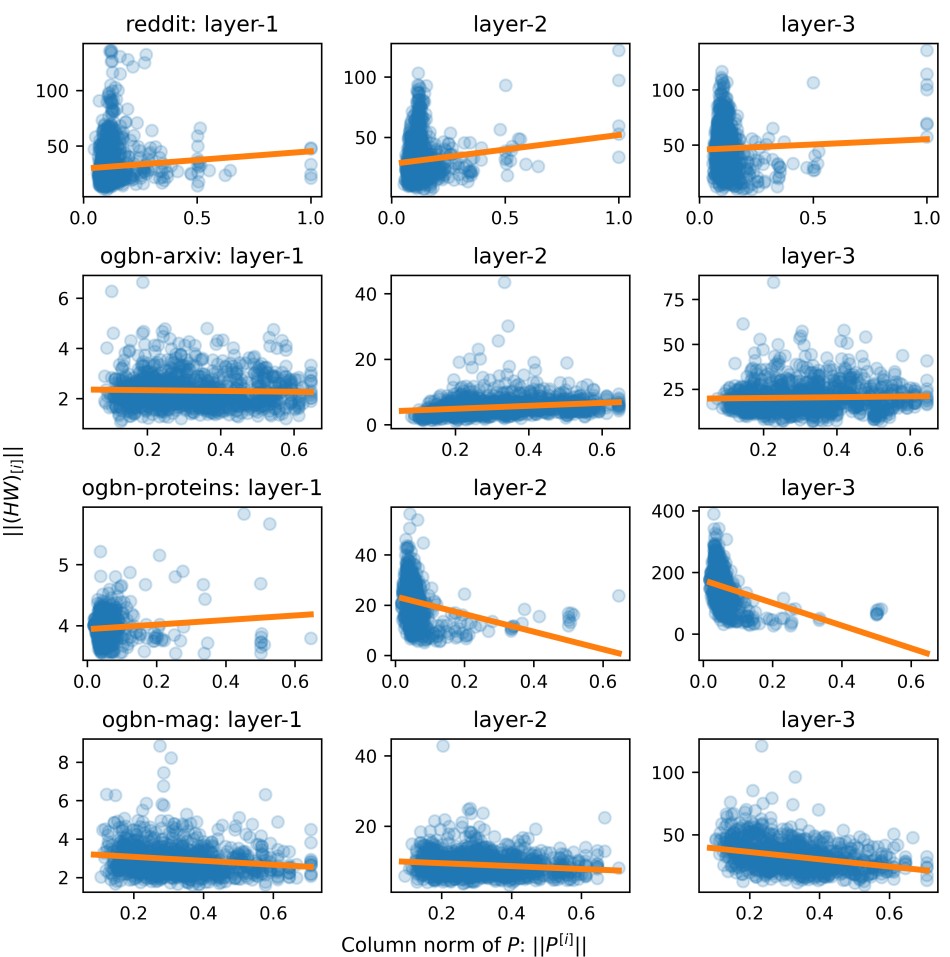

Figure 6: Regression of $\|(\boldsymbol{HW})_{[i]}\| \sim \beta_0 + \beta_1\|\boldsymbol{P}^{[i]}\|$ on Reddit, ogbn-arxiv, ogbn-proteins, and ogbn-mag datasets. 3-layer GCN is trained by full-bacth sampler. The fitted regression line is in orange color.

## D.2 FastGCN/FastGCN+debiasing

In this subsection, we present the regression analysis for FastGCN/FastGCN+debiasing in this subsection. The regression results are illustrated in Figures 7 and 8; more details can be found in Table 9. The distribution patterns of the $(\|(\boldsymbol{H}\boldsymbol{W})_{[i]}\|, \|\boldsymbol{P}^{[i]}\|)$ pairs in FastGCN/FastGCN+debiasing are similar to the patterns in Figures 6 and 9; we can analogously draw the conclusion that for models trained by FastGCN/FastGCN+debiasing, the assumption $\|(\boldsymbol{H}\boldsymbol{W})_{[i]}\| \propto \|\boldsymbol{P}^{[i]}\|$ tends to not hold.

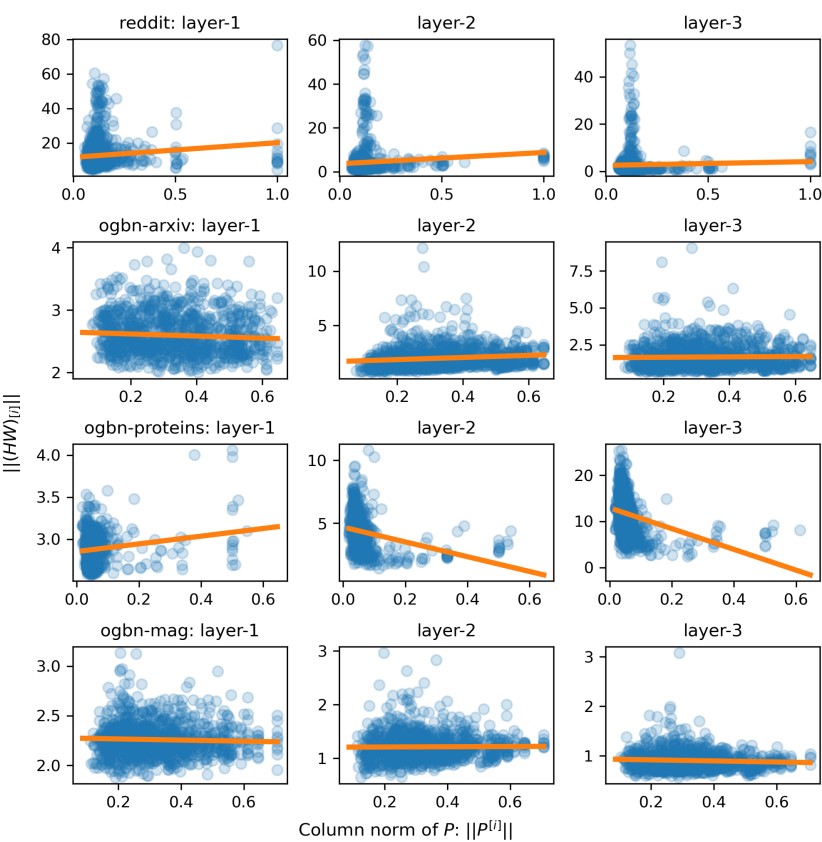

Figure 7: Regression of $\|(\boldsymbol{H}\boldsymbol{W})_{[i]}\| \sim \beta_0 + \beta_1\|\boldsymbol{P}^{[i]}\|$ on Reddit, ogbn-arxiv, ogbn-proteins, and ogbn-mag datasets. 3-layer GCN is trained by FastGCN sampler. The fitted regression line is in orange color.

Table 9: Regression coefficients for $\|(\boldsymbol{H}\boldsymbol{W})_{[i]}\| \sim \beta_0 + \beta_1\|\boldsymbol{P}^{[i]}\|$. The data come from 3-layer GCNs trained with FastGCN/FastGCN+d(ebiasing) respectively. No regression has high $R^2$ and the $R^2$ for positive $\beta_1$'s are highlighted in boldface.

| Method | Dataset | Layer 1 | | | Layer 2 | | | Layer 3 | | |
|---|---|---|---|---|---|---|---|---|---|---|
| | | $\beta_0$ | $\beta_1$ | $R^2$ | $\beta_0$ | $\beta_1$ | $R^2$ | $\beta_0$ | $\beta_1$ | $R^2$ |
| FastGCN | ogbn-arxiv | 2.651 | -0.165 | 0.005 | 1.627 | 0.981 | **0.017** | 1.640 | 0.123 | **<0.001** |
| | reddit | 11.773 | 8.453 | **0.009** | 3.639 | 5.155 | **0.006** | 2.524 | 1.587 | **0.001** |
| | ogbn-proteins | 2.853 | 0.457 | **0.022** | 4.692 | -5.883 | 0.069 | 12.938 | -22.552 | 0.098 |
| | ogbn-mag | 2.278 | -0.057 | 0.001 | 1.208 | 0.023 | **<0.001** | 0.938 | -0.108 | 0.005 |
| FastGCN+d | ogbn-arxiv | 2.847 | -0.164 | 0.004 | 1.984 | 1.277 | **0.021** | 2.385 | 0.111 | **<0.001** |
| | reddit | 13.747 | 9.712 | **0.008** | 4.654 | 6.322 | **0.007** | 3.264 | 1.563 | **0.001** |
| | ogbn-proteins | 3.090 | 0.397 | **0.017** | 5.372 | -7.357 | 0.072 | 11.395 | -20.179 | 0.102 |
| | ogbn-mag | 2.310 | -0.067 | 0.002 | 1.194 | -0.009 | <0.001 | 0.999 | -0.147 | 0.006 |

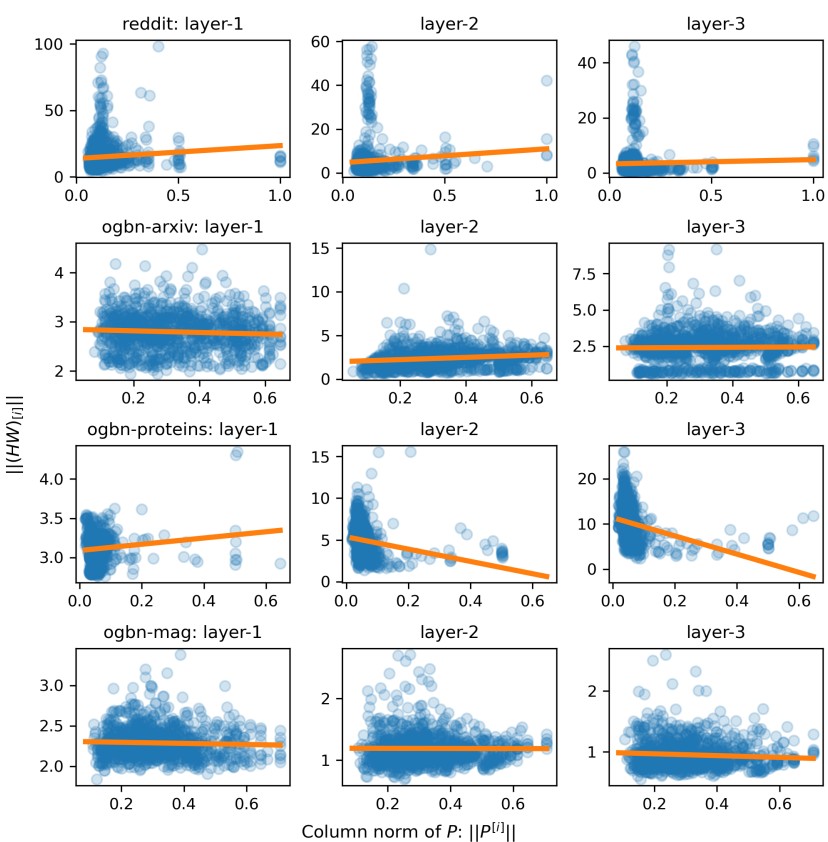

Figure 8: Regression of $\|(\boldsymbol{HW})_{[i]}\| \sim \beta_0 + \beta_1 \|\boldsymbol{P}^{[i]}\|$ on Reddit, ogbn-arxiv, ogbn-proteins, and ogbn-mag datasets. 3-layer GCN is trained by FastGCN+debiasing sampler. The fitted regression line is in orange color.

### D.3   LADIES: setting 1

This subsection presents regression plots for $\|(\boldsymbol{HW})_{[i]}\| \sim \|\boldsymbol{P}^{[i]}\|$ as supplementary to Figure 1. Note that a different regression setting $\|(\boldsymbol{HW})_{[i]}\| \sim \|\boldsymbol{QP}^{[i]}\|$ is collected in Appendix D.4.

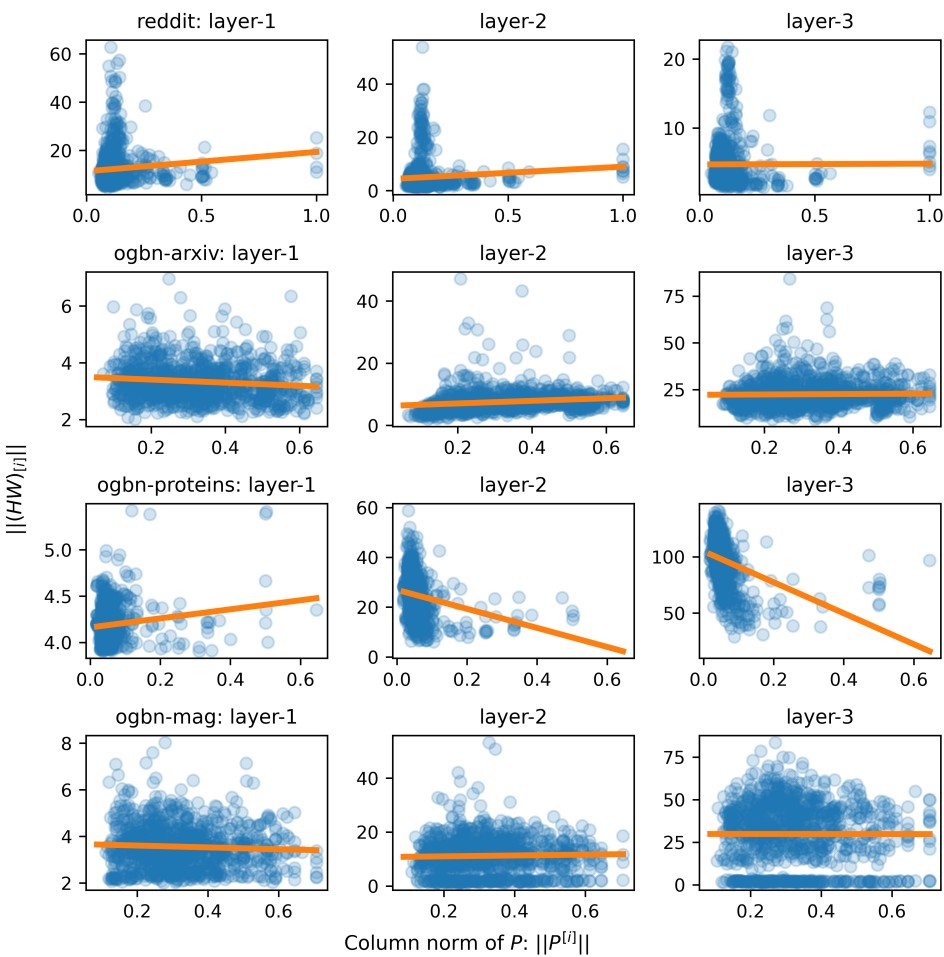

Figure 9: Regression of $\|(\boldsymbol{HW})_{[i]}\| \sim \beta_0 + \beta_1 \|\boldsymbol{P}^{[i]}\|$ on Reddit, ogbn-arxiv, ogbn-proteins, and ogbn-mag datasets. 3-layer GCN is trained by LADIES. The fitted regression line is in orange color.

### D.4   LADIES: setting 2

As remarked in the footnote in Section 4.1, we view the assumption $\|(\boldsymbol{HW})_{[i]}\| \propto \|\boldsymbol{QP}^{[i]}\|$ in LADIES as a randomized version of the one $\|(\boldsymbol{HW})_{[i]}\| \propto \|\boldsymbol{P}^{[i]}\|$ and hence focus on the latter regression setting. However, it may still be interesting to present regression analysis of $\|(\boldsymbol{HW})_{[i]}\| \sim \|\boldsymbol{QP}^{[i]}\|$ in this subsection to study the corresponding assumption $\|(\boldsymbol{HW})_{[i]}\| \propto \|\boldsymbol{QP}^{[i]}\|$ in LADIES. Compared to the original assumption in FastGCN, setting $\|\boldsymbol{QP}^{[i]}\|$'s as the predictor causes some different patterns in the regression.

- There are more empty columns in $\boldsymbol{QP}$ than in $\boldsymbol{P}$. For a single selection matrix, we have fewer $(x, y)$ pairs. To compensate, we pick 500 $\boldsymbol{Q}$'s and record non-zero $\|\boldsymbol{QP}^{[i]}\|$'s and corresponding $\|(\boldsymbol{HW})_{[i]}\|$'s as the regression input.

- There is also a higher portion of high-leverage points after considering the selection matrix $\boldsymbol{Q}$—the points with large $\|\boldsymbol{Q}\boldsymbol{P}^{[i]}\|$'s are fewer while they have higher influence on the coefficients (which is not favored in regression analysis since they increase the standard error of the estimated coefficients).

The regression results are illustrated in Figures 10 and 11; more details can be found in Table 10. We note the regression results still fail to support the proportionality assumption in LADIES: most $\beta_1$'s are negative, and even for the positive $\beta_1$'s the $R^2$ (the coefficient of determination, equal to the square of the correlation coefficient in univariate linear regression) is small, which matches the observation in the figures that there is no clear proportionality in the data.

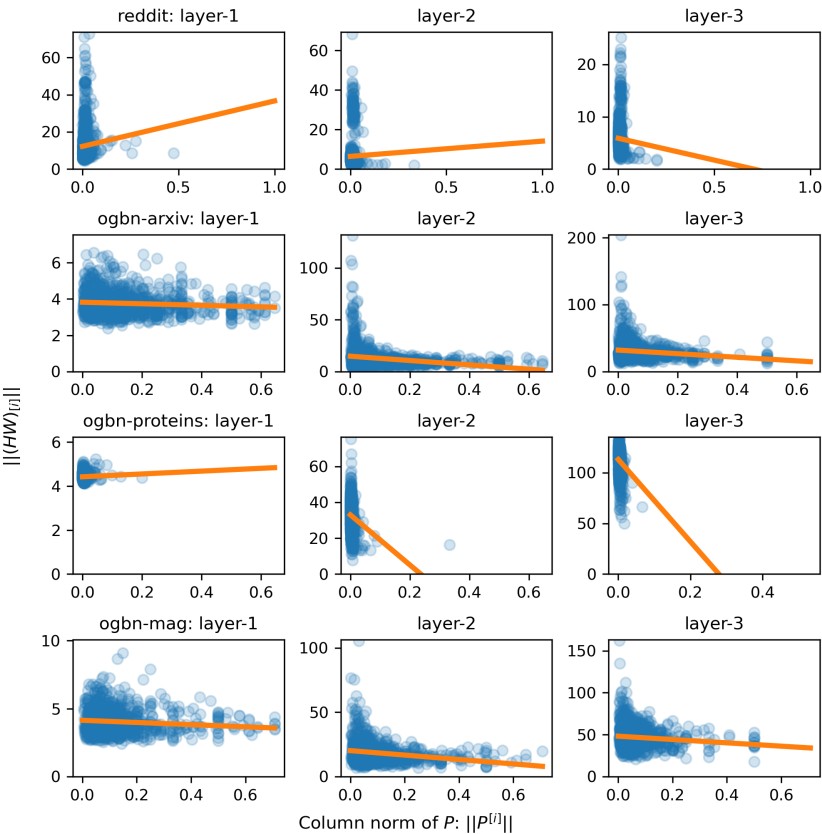

Figure 10: Regression of $\|(\boldsymbol{HW})_{[i]}\| \sim \beta_0 + \beta_1 \|\boldsymbol{Q}\boldsymbol{P}^{[i]}\|$ on Reddit, ogbn-arxiv, ogbn-proteins, and ogbn-mag datasets. 3-layer GCN is trained by LADIES sampler. The fitted regression line is in orange color.

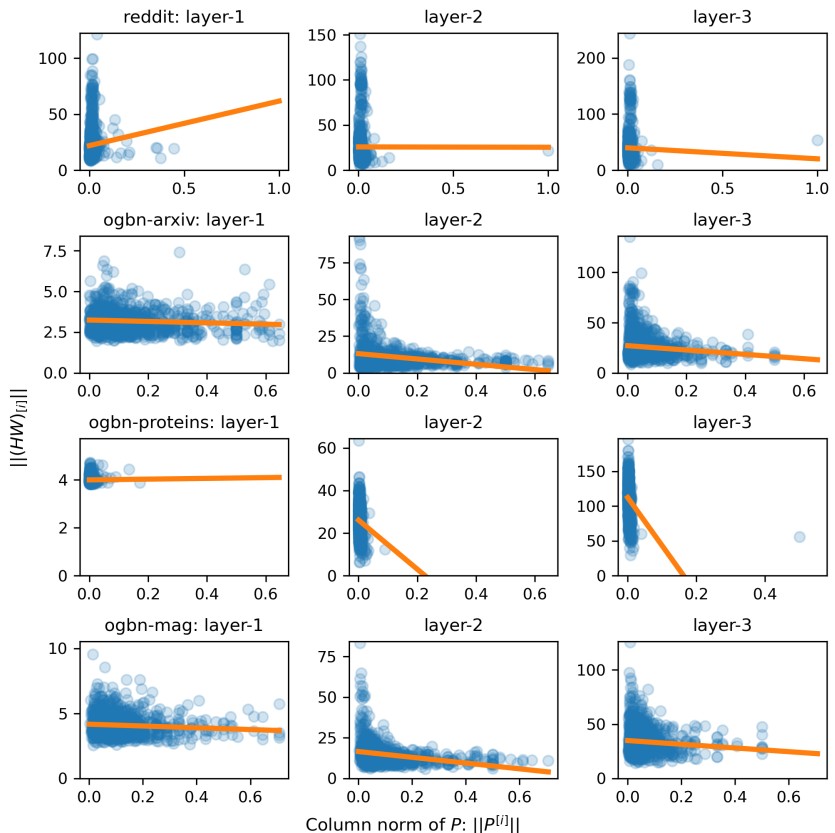

Figure 11: Regression of $\|(\boldsymbol{HW})_{[i]}\| \sim \beta_0 + \beta_1\|\boldsymbol{QP}^{[i]}\|$ on Reddit, ogbn-arxiv, ogbn-proteins, and ogbn-mag datasets. 3-layer GCN is trained by LADIES+debiasing sampler. The fitted regression line is in orange color.

Table 10: Regression coefficients for $\|(\boldsymbol{HW})_{[i]}\| \sim \beta_0 + \beta_1\|\boldsymbol{QP}^{[i]}\|$. The data come from 3-layer GCNs trained with LADIES/LADIES+d(ebiasing) respectively. No regression has high $R^2$ and the $R^2$ for positive $\beta_1$ are highlighted in boldface.

| Method | Dataset | Layer 1 | | | Layer 2 | | | Layer 3 | | |
|---|---|---|---|---|---|---|---|---|---|---|
| | | $\beta_0$ | $\beta_1$ | $R^2$ | $\beta_0$ | $\beta_1$ | $R^2$ | $\beta_0$ | $\beta_1$ | $R^2$ |
| LADIES | ogbn-arxiv | 3.819 | -0.424 | 0.007 | 14.827 | -20.853 | 0.048 | 32.079 | -26.803 | 0.028 |
| | reddit | 12.218 | 24.459 | **0.006** | 6.409 | 7.711 | **0.001** | 5.897 | -8.366 | 0.002 |
| | ogbn-proteins | 4.425 | 0.642 | **0.006** | 33.042 | -139.554 | 0.033 | 113.204 | -405.534 | 0.060 |
| | ogbn-mag | 4.138 | -0.797 | 0.011 | 20.193 | -17.247 | 0.039 | 48.205 | -19.879 | 0.011 |
| LADIES+d | ogbn-arxiv | 3.593 | -0.538 | 0.011 | 6.503 | 4.125 | **0.025** | 19.388 | 1.374 | **0.001** |
| | reddit | 22.122 | 13.941 | **0.008** | 21.783 | 13.231 | **0.004** | 37.739 | 1.320 | **<0.001** |
| | ogbn-proteins | 4.025 | 0.296 | **0.012** | 24.889 | -36.718 | 0.080 | 109.424 | -197.488 | 0.152 |
| | ogbn-mag | 4.184 | -0.653 | 0.009 | 12.524 | -0.920 | 0.001 | 31.837 | 0.118 | **<0.001** |

