# OpenReview forum: "Calibrate and Debias Layer-wise Sampling for Graph Convolutional Networks"
_TMLR — Accepted by TMLR_

### Review · Reviewer_31rs · 2022-08-11

**Summary Of Contributions:**

This paper aims to improve layer-wise sampling for graph convolutional networks (GCN), with a particular focus on two popular models, i.e., FastGCN and LADIES. The authors pointed out two shortcomings of the current sampling methods: 1) the sampling probabilities are suboptimal because they are often constructed under unguaranteed assumptions; 2) the sampling is performed without replacement, bringing in unexpected estimation biases. Accordingly, the authors proposed two methods to address these issues respectively. They propose to generate sampling probabilities following the Principle of Maximum Entropy and introduce an iterative debiasing algorithm. Results show that the proposed methods could reduce the matrix approximation error and improve downstream performance.

**Broader Impact Concerns:**

I didn't find serious ethical problems.

**Requested Changes:**




1) The analysis for the sampling probability problem in Section 4 is not convincing

    a)	First of all, it is FastGCN that assumes $||(\mathbf{H}\mathbf{W})_{\[i\]}||\propto ||\mathbf{P}^{\[i\]}||$ rather than LADIES. Why do the authors examine the correlation between them for LADIES? For LADIES, it should be $||\mathbf{Q}\mathbf{P}^{\[i\]}||$. Such a mismatch makes the results in Table 2 and Figure 1 less convincing, so it’s still questionable whether the traditional assumption really deteriorates the sampling probability.

    b)	Secondly, stating that “little information of $||(\mathbf{H}\mathbf{W})_{\[i\]}||$ can be explicitly retrieved from $\mathbf{P}$” is annoying. Both position and negative coefficients could tell that these two variables are correlated, albeit with a different relationship. Instead, by using your proposed approach dropping the approximated part or assuming a constant of 1 for $||\mathbf{P}^{\[i\]}||$, the correlation is totally ignored.

    c)	Thirdly, from intuition, the proposed sampling method should perform better in settings where negative coefficients are observed but worse in those positive-coefficient setups. But this is not true. In Table 4, the “flat” method performs consistently better than the LADIES baseline on Reddit (where LADIES show positive coefficients across all layers). Could you explain this observation? This may be related to the mismatch problem in a).

    d)	The debiasing algorithm introduced in Section 5 shows promising performance in Table 4. Would fixing the bias problem also alleviates the probability issue? What about the coefficients for “LADIES/FastGCN + debiasing”?

2) The identified problems are assumed to be general to layer-wise sampling based GCN, but the authors only performed a full examination with LADIES. Since the authors also focused on FastGCN in this paper, it makes a lot of sense to provide full results for FastGCN as well, which could offer great insights to the readers on whether the findings in this paper are generalizable.


**Strengths And Weaknesses:**

Strengthens:

* This paper points out two problems related to the traditional sampling strategy used in layer-wise sampling for GCN; the problems might be interesting to the GCN research community.
* The proposed methods are simple and obtained encouraging performance as reported in the paper.

Weaknesses:

* Some motivational analyses and statements are not convincing.
* More empirical evidence should be given to show the generality of the identified problems.

---

> ### Author Response · Authors · 2022-08-12
> **Thank You to Reviewer 31rs**
>
> We sincerely thank Reviewer 31rs for providing insightful reviews and valuable comments. Specifically, we will accordingly do more analysis of FastGCN to verify whether our claim is general enough; before we paid more attention to LADIES simply due to its better performance.
>
> Another post will be made after we finish the analysis and revise the paper based on the suggested changes and the other reviewers' comments. We plan to upload the revision in two weeks.

---

### Review · Reviewer_LYct · 2022-10-16

**Summary Of Contributions:**

The paper investigates two issues on layer-wise sampling schemes for GCNs (FastGCN and LADIES). The first one refers to wrong assumptions on sampling probabilities, which is tackled by adopting the Maximum Entropy principle. The second issue refers to biased estimation due to sampling without replacement, for which the paper introduces a debiasing sequential procedure. Experiments on relevant benchmarks show better convergence and accuracy of LADIES combined with the proposed solutions.

**Broader Impact Concerns:**

I have no broader impact concerns.

**Requested Changes:**

Overall, this is a solid and interesting work. I only have a few minor requests/comments:

- Adding to the captions whether time was measured on GPU or CPU (like in Table 3) would be helpful. Also, when the paper says "our procedure can be performed independently on CPU, it will not retard the training on GPU", it neglects CPU-GPU communication time. Is this communication time insignificant here? Somehow I would expect the sequential debiasing procedure on GPU to cause a higher effect on the overall training time.
- It would be interesting to report accuracy numbers for FastGCN+flat+debias in the Appendix.
- Saying that the success of GCNs comes from the "successful approximation to the spectral graph convolutions" (Sec. 1.1) is questionable. For instance, I would simply begin with "GCN has achieved great success ...".
- The paper would benefit from a textual review. Here are some typos:
  - Sparese (Page 2)
  - Matrix \hat{D} should be bold. (Page 2)
  - Use \mathcal{S} (before equation 2)
  - We insist ON sampling ... (Remark)


**Strengths And Weaknesses:**

**Strengths**
- The paper identifies an important theory-practice gap regarding assumptions and biases in layer-wise sampling on GCNs.
- The proposed solutions are simple and address the identified issues.
- Experiments include relevant large-scale benchmarks and validate the efficacy of the proposed solutions.

**Weaknesses**
- I am not sure if this contribution advances the state-of-the-art in (large-scale) node classification since vanilla GCNs are weak models, usually outperformed by simple fast approaches [e.g.,1, 2].
- The experimental setup only considers shallow models (up to 3 layers).
- Except for Table 3, it is unclear how time is measured (e.g., if it was measured on GPU or CPU). Reporting only CPU numbers could be misleading.

[1] https://arxiv.org/abs/2010.13993

[2] https://arxiv.org/abs/1902.07153

---

> ### Author Response · Authors · 2022-10-16
> **Thank You to Reviewer LYct**
>
> We appreciate the helpful comments from Reviewer LYct. We will accordingly revise our paper and make another reply after all the reviews are out.

---

### Review · Reviewer_9G1C · 2022-10-26

**Summary Of Contributions:**

In this paper, the authors points out two potential drawbacks (sub-optimal sampling probabilities and sampling without replacement) in the common practice for layer-wise sampling of graph convolutional networks. To address these, thr authors 1) propose a more conservative principle to construct importance sampling probabilities relying on the Principle of Maximum Entropy; 2) suggest a debiasing algorithm to deal with the bias induced by sampling without replacement.

**Broader Impact Concerns:**

No.

**Requested Changes:**

Please see the weakness part above.

**Strengths And Weaknesses:**

Strengths:
1.	The authors provide in-depth analysis on the two drawbacks in layer-wise sampling and the results are convincing.
2.	The proposed sampling probabilities (Eq (7)) is improved based on the violated assumpation in FastGCN/LADIES, which in spirit is interesting.
3.	It seems that the proposed debiasing method can be further adapted to more general machine learning tasks involving importance sampling without replacement, which may broden the application areas.

Weaknesses&Questions:
Though the overall paper is interesting, I have the following concerns:
1.	From Table 4, when choosing LADIES(2) as the baseline, the training time increased when additionally applying “flat” on the “debias” variant (21 vs 14). However, when choosing LADIES as the baseline, the training time decreases from “w/ debias”(19) to “w/ flat&debias”(14). Could you provide some explanations?
2.	In terms of some metrics (e.g., ogbn-arxiv), the performance decreases when additionally applying “flat” on the “debias” variant. From these point of view, the proposed two strategies may be conflict in some application sceneries..
3.	From Figure2, the “flat&debiased” baseline has similar performance compared with the “debiased” baseline when the number of sampled nodes is relatively small (around 1000). Could the two strategies be automatically selected via some threasholds to achieve a bettern trade-off between effectiveness and efficiency.

---

### Decision · Action_Editors · 2023-01-09

**Recommendation:** Accept with minor revision

**Comment:**

The paper aims to improve efficiency of graph neural networks. The paper focuses on improving sampling strategy for graph convolution networks and in this regards, the author begin by identifying shortcomings on current methods like LADIES or FastGCN, like bias caused by sampling without replacement. A new sampling distribution with flattening and a debiased sampling algorithm is proposed. Strong empirical improvements is achieved across several large-scale benchmarks when correcting LADIES approach, but unfortunately the proposed method seems to not work well with FastGCN. Nevertheless, the proposed method and technique are correct, experimental result demonstrate effectiveness in one way, and the approach will be of interest to the community, hence I propose to accept the paper with some minor modifications.

1. Add limitation about proposed method not working for FastGCN. It can't be just sparsity of FastGCN is the cause as adding the debiased sampling almost always hurts FastGCN's performance.
2. As pointed out by 31rs, it would be instructive for readers to write about why proposed method with flattening still works better than LADIES on datasets where proportionality assumption holds? Is it better hyper-parameter tuning?
3. Rename appendix D.3, it can't be titled author response.

I will thank the authors for the patience and all the updates for more clarity and thorough evaluation.

**Audience:**

Yes, both graph neural network and efficient ML community will be interesting in this paper.

**Claims And Evidence:**

Yes